# Motor processivity and speed determine structure and dynamics of microtubule-motor assemblies

**Rachel A Banks[1], Vahe Galstyan[1], Heun Jin Lee[2], Soichi Hirokawa[2], Athena Ierokomos[3], Tyler D Ross[4], Zev Bryant[5], Matt Thomson[1], Rob Phillips[1,2,6]\***

[1]Division of Biology and Biological Engineering, California Institute of Technology, Pasadena, United States; [2]Department of Applied Physics, California Institute of Technology, Pasadena, United States; [3]Biophysics Program, Stanford University, Stanford, United States; [4]Department of Computing and Mathematical Science, California Institute of Technology, Pasadena, United States; [5]Department of Bioengineering, Stanford University, Stanford, United States; [6]Department of Physics, California Institute of Technology, Pasadena, United States

**\*For correspondence:**
phillips@pboc.caltech.edu

**Competing interest:** The authors declare that no competing interests exist.

**Abstract** Active matter systems can generate highly ordered structures, avoiding equilibrium through the consumption of energy by individual constituents. How the microscopic parameters that characterize the active agents are translated to the observed mesoscopic properties of the assembly has remained an open question. These active systems are prevalent in living matter; for example, in cells, the cytoskeleton is organized into structures such as the mitotic spindle through the coordinated activity of many motor proteins walking along microtubules. Here, we investigate how the microscopic motor-microtubule interactions affect the coherent structures formed in a reconstituted motor-microtubule system. This question is of deeper evolutionary significance as we suspect motor and microtubule type contribute to the shape and size of resulting structures. We explore key parameters experimentally and theoretically, using a variety of motors with different speeds, processivities, and directionalities. We demonstrate that aster size depends on the motor used to create the aster, and develop a model for the distribution of motors and microtubules in steady-state asters that depends on parameters related to motor speed and processivity. Further, we show that network contraction rates scale linearly with the single-motor speed in quasi-one-dimensional contraction experiments. In all, this theoretical and experimental work helps elucidate how microscopic motor properties are translated to the much larger scale of collective motor-microtubule assemblies.

## Editor's evaluation

Banks et al. demonstrate that the organization and dynamics of microtubule/kinesin asters depend upon the speed and processivity of motors. By combining in vitro reconstitutions with theory, they are able to extract parameters that relate to the dynamics of the motors. This study is of interest to readers working on microtubules, motors, and in the active matter physics field.

## Introduction

A signature feature of living organisms is their ability to create beautiful, complex patterns of activity, as exemplified in settings as diverse as the famed flocks of starlings in Rome or the symmetrical and dazzling microtubule arrays that separate chromosomes in dividing cells (*Popkin, 2016*). While such organization in nature has long captured the attention of artists and scientists alike, many questions

remain about how the patterns and structures created by living organisms arise. In active systems such as bird flocks or microtubule-motor arrays, energy is consumed at the local level of the individual actors, and constituents move based on interactions with their neighbors. These local actions create patterns at scales hundreds to thousands of times larger than the individual constituent. How the specific microscopic activity of each individual leads to the final large-scale assembly formed remains an open question in these systems from the molecular to organismal level.

The motor-microtubule system is an excellent system in which to test this question, as many motor proteins with a variety of properties, such as speeds, stall and detachment forces, processivities, and directionalities, exist in nature. These motors play a variety of roles in cells; some transport cargo while others localize to distinct regions of the mitotic spindle (*Wickstead and Gull, 2006*; *Mann and Wadsworth, 2019*; *Endow et al., 1994*; *Pavin and Tolić, 2021*; *Anjur-Dietrich et al., 2021*). Studies have investigated how the microscopic properties of these motors makes them uniquely suited to their cellular role. For example, kinesin-1s high speed and processivity make it excellent at transporting cargo (*Grover et al., 2016*; *Hirokawa et al., 1991*). However, in in vitro systems, kinesin-1 tetramers are able to form asters, extensile networks, and contractile networks (*Surrey et al., 2001*; *Sanchez et al., 2012*; *DeCamp et al., 2015*; *Ross et al., 2019*). Ncd (kinesin-14) and Kif11 (kinesin-5) have similarly been shown to form asters in vitro, yet it remains unclear how the properties of these motors affect the structure and dynamics of the assemblies created (*Surrey et al., 2001*; *Sanchez et al., 2012*; *DeCamp et al., 2015*; *Ross et al., 2019*; *Roostalu et al., 2018*).

In this work, we create motor-microtubule structures with a variety of motors and develop theoretical models to connect the motor properties to the organization and dynamics of the assemblies. Our recently developed optogenetic in vitro motor-microtubule system demonstrated the formation of asters and other contractile networks with kinesin-1 (K401) upon light activation (*Ross et al., 2019*). Briefly, we fused K401 motors to an optogenetic pair of light-dimerizable proteins, such that in the presence of light the optogenetic pair bind, acting as a crosslink between microtubules that the motor heads are walking along. Previously, we showed that this scheme enabled us to form microtubule structures with spatiotemporal control by illuminating regions of the sample at will. We now show how this system can be re-purposed to ask a new set of questions with kinesin-5 (Kif11) and kinesin-14 (Ncd), and form asters of varying sizes with each motor, demonstrating light-controlled aster formation with these motors for the first time. Our controlled structure formation with these various motors enabled us to develop a theoretical model connecting the distribution of motors and microtubules in asters that depends on microscopic motor properties. We find that calculated motor distributions in an aster depend on the motor properties and fit with our experimental data. Further, by using motors with different speeds, we find that contraction rates in quasi-one-dimensional microtubule networks directly depend on the single-motor velocity. This theoretical and experimental work sheds light on the ways that microscopic motor properties are reflected in the 1000-fold larger length scale of motor-microtubule assemblies.

## Results

### Aster size depends on motor used

We build on the foundational work that demonstrated the ability to control motor-microtubule systems with light (*Ross et al., 2019*) to consider a new set of motors with different fundamental properties. In brief, kinesin motors are fused to the light-dimerizable pair iLid and micro. In the absence of light, motor dimers walk along microtubules but do not organize them; upon activation with light, the motor dimers couple together to form tetramers, crosslinking the microtubules they are walking along as shown in *Figure 1A*. The optogenetic bond lasts for about 20 s before reverting to the undimerized state, thus in our experiments, we repeatedly illuminate the sample every 20 s (*Guntas et al., 2015*). As demonstrated by Ross et al., projecting a cylinder of light on the sample results in the formation of an aster, and different structures can be formed and manipulated by illumination with different light patterns. For the purposes of this study, we were careful to remain in a regime of motor and microtubule concentrations that produced a single aster upon illumination. However, by varying concentrations of the motors and microtubules, it is possible to form multiple smaller asters within the region, a few examples of which are shown in *Figure 1—figure supplement 1*. How varying the composition of the reaction mixture impacts the resulting structures warrants further investigation.

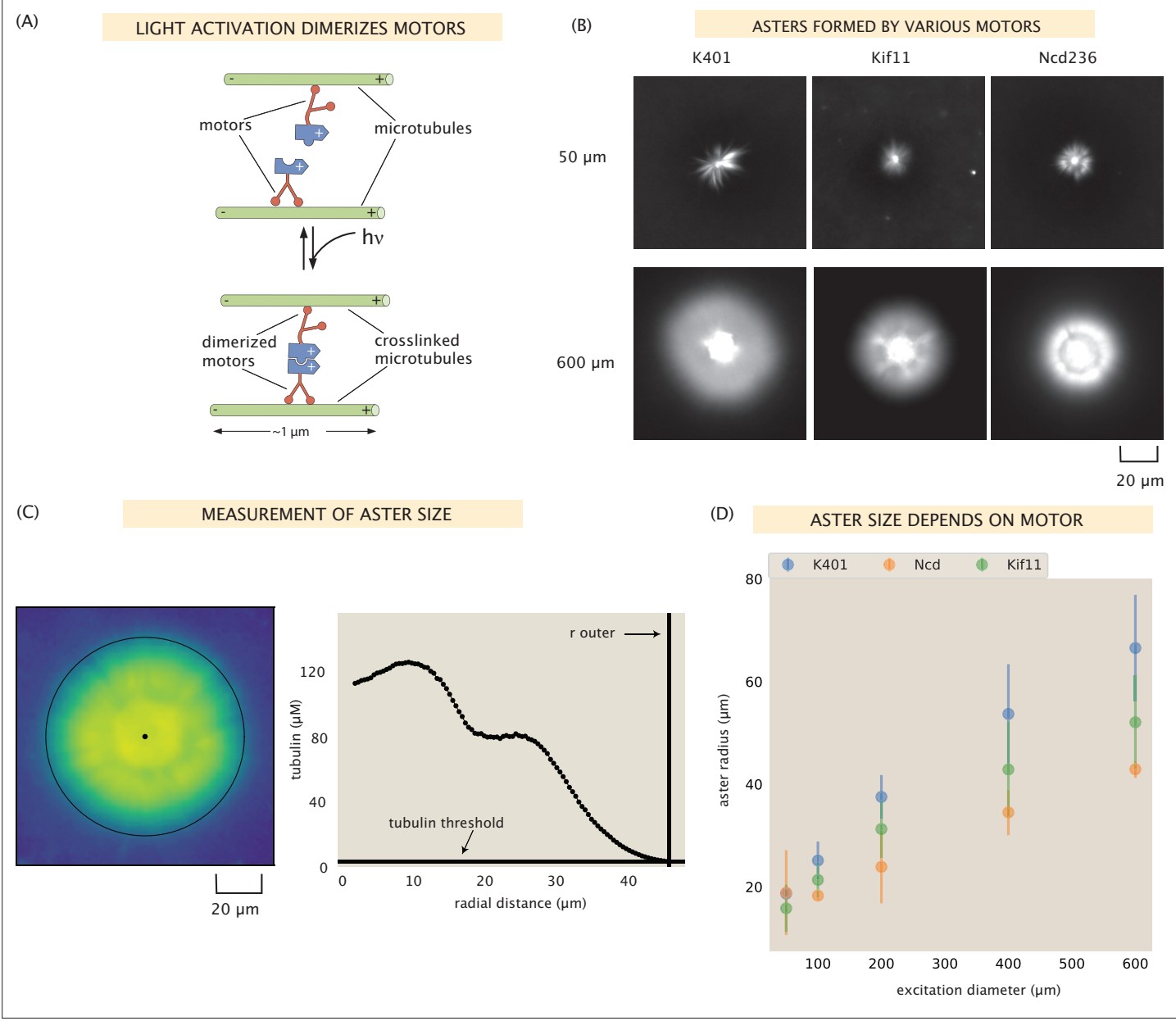

**Figure 1.** Aster size depends on motor used. (**A**) Motor heads are fused to optogenetic proteins such that activation with light causes the formation of motor tetramers (dimer of dimers). Motors are shown walking toward the microtubule plus-end. K401 and Kif11 walk in this direction, however Ncd is minus-end directed. (**B**) Images of the microtubule fluorescence for asters formed with each of the motors excited with a disk either 50 or 600 μm in diameter. (**C**) Image of the microtubule fluorescence from an aster with the measured size represented with the outer black circle. The plot on the right shows the radial microtubule concentration as revealed by fluorescence intensity; the threshold concentration used to determine the aster size is shown as a black horizontal line. (**D**) Mean aster size ($n \approx 5$ asters for each condition) for the three motors and different excitation diameters; the error bars represent the standard deviation.

The online version of this article includes the following figure supplement(s) for figure 1:

**Figure supplement 1.** Examples of multiple asters formed.

**Figure supplement 2.** Example images of microtubule fluorescence of asters made with each motor used and each excitation diameter.

**Table 1.** Properties of motor proteins used in this study.

Speeds were measured by us by gliding assay, the processivity and directionality are literature values.

| Motor | Speed | Processivity | Direction |
|---|---|---|---|
| K401 | ≈ 600 nm/s | ≈100 steps (***Belmonte et al., 2017***) | Plus (***Hirokawa et al., 1991***) |
| Ncd236 | ≈115 nm/s | Not processive (***Hentrich and Surrey, 2010***, ***Nédélec et al., 2001***, ***Lee and Kardar, 2001***) | Minus (***Sankararaman et al., 2004***) |
| Kif11(513) | ≈70 nm/s | ≈10 steps (***Aranson and Tsimring, 2006***) | Plus (***Aranson and Tsimring, 2006***) |

In this work, we aim to determine how the properties of the motor affect the resulting structures. While experiments with this system were previously performed with *Drosophila melanogaster* kinesin-1 motors (K401) (***Ross et al., 2019***), in the present work, we investigate if other kinesin motor species with different intrinsic properties such as speed and processivity would lead to light-inducded microtubule organization. Toward this end, we use the same light-dimerizable scheme to form microtubule structures with two other motors: Ncd (*D. melanogaster* kinesin-14) and Kif11 (*Homo sapiens* kinesin-5). The single-molecule properties of all three motors we use are summarized in ***Table 1***. We measure the speed of each motor species by gliding assays (SI section 'Gliding assay'); the processivities are based on literature values. Further, we fluorescently label the motors using mVenus or mCherry to visualize the motors and microtubules in separate imaging channels within the same assay (see ***Supplementary file 1***).

As seen in ***Figure 1B*** and ***Figure 1—figure supplement 2***, each of these motors is able to form asters of varying sizes in our system. It was previously unclear whether there were limits to speed, processivity, or stall force that might prevent any of these motors from forming asters in our light-controlled system, although Ncd has previously been shown to form asters as constitutive oligomers (***Belmonte et al., 2017***; ***Hentrich and Surrey, 2010***). We found that all were able to form asters upon illumination by various excitation diameters ranging from 50 to 600 μm. Interestingly, the dynamics of aster formation by these motors seemed to roughly scale with the motor speeds – K401 formed asters the quickest, followed by Ncd236, and Kif11 took the most time to form an aster.

We sought to determine if there are discernible differences between the asters formed with the various motors. First, we measured the size of the asters using the distribution of fluorescently labeled microtubules, which peaks in the center of the aster and generally decreases moving outward, as shown in ***Figure 1C***. We tend to observe a shoulder in the microtubule distribution (around 20 in the example in ***Figure 1C***). This is around the size of the disordered aster core, which is discussed in SI section 'Disordered aster core'. We defined the outer radius of the aster as the radius at which the microtubule fluorescence is twice the background microtubule concentration (see ***Figure 1C*** for an example aster outer radius determination). We found that this method agreed well with a visual inspection of the asters (***Figure 1—figure supplement 2***).

We found that aster size increases with excitation diameter, as shown in ***Figure 1B, D***, consistent with what was shown by Ross et al. for K401 (***Ross et al., 2019***). In Ross et al., it was determined that the aster size roughly scaled with the volume of the excitation area, suggesting that the number of microtubules limits the size of the aster. This hints that there may be a density limit to the microtubules in an aster. Interestingly, we find that the size of the asters also depends on the motor used. For each excitation diameter, except for the 50 μm case, K401 formed the largest asters and Ncd formed the smallest, with Kif11 producing asters of intermediate size (***Figure 1D***). What is it about the different motors that confers these different structural outcomes? We found that this trend correlates with motor processivity; K401 is the most processive, followed by Kif11, and then Ncd. This is similar to the findings in ***Surrey et al., 2001***, in which intensity of aster formation was related to motor processivity. Other factors could also be contributing to aster size such as the ratio of microtubules to motors as was suggested by ***Surrey et al., 2001***, but not investigated in the present work, or the motor stall force.

## Spatial distribution of motors in asters

The nonuniform distribution of filaments and motors in an aster is a key feature of its organization and has been the subject of previous studies. In these studies, continuum models were developed for motor-filament mixtures which predicted the radial profile of motors in confined two-dimensional systems (*Nédélec et al., 2001*; *Lee and Kardar, 2001*; *Sankararaman et al., 2004*; *Aranson and Tsimring, 2006*). A notable example is the power-law decay prediction by Ndlec et al., who obtained it for a prescribed organization of microtubules obeying a $1/r$ decay law (*Nédélec et al., 2001*). Measuring the motor profiles in asters formed in a quasi-two-dimensional geometry (with the $z$-dimension of the sample being only a few microns deep) and fitting them to a power-law decay, the authors found a reasonable yet noisy match between the predicted and measured trends in the decay exponent.

In our work, we also develop and test a minimal model that predicts the motor profile from the microtubule distribution and the microscopic properties of the motor. Our well-defined asters of various sizes shown above, created with varying motor properties yields an opportunity for us to test how our model and assess how the microscopic motor properties are translated to the aster scale. Additionally, in contrast to the earlier study (*Nédélec et al., 2001*), asters formed in our experiments are three-dimensional due to the much larger depth of the flow cells (roughly 100 μm), and are thus more similar to the structures observed in cells. While the largest asters we form are likely partially compressed in the $z$-direction, we assume that this effect does not significantly alter the protein distributions in the central $z$-slice. For modeling purposes, then, we consider our asters to be radially symmetric outside the central disordered region (which we refer to as the aster core, as depicted schematically in *Figure 2A*). The core has a typical radius of $\approx 15$ μm, beyond which microtubules have a predominantly polar organization (see SI section 'Disordered aster core' for the discussion of the two aster regions and an example Pol-Scope image that demonstrates their distinction).

Similar to the treatment in earlier works (*Nédélec et al., 2001*; *Sankararaman et al., 2004*; *Aranson and Tsimring, 2006*), we introduce two states of the motor – an unbound state where the motor can freely diffuse with a diffusion constant $D$ and a bound state where the motor walks toward the aster center with a speed $v$, depicted in *Figure 2B*. In the steady state of the system, which we assume our asters have reached at the end of the experiment, microtubules on average have no radial movement and hence, do not contribute to motor speed. To assess the validity of this assumption, we performed fluorescence recovery after photobleaching (FRAP) experiments of steady-state asters, and observe little radial flux of the microtubules, an example is shown in *Appendix 1—figure 4*. They are still dynamic, as can be seen by the angular motion that leads to the recovery of fluorescence in the photobleached areas. We denote the rates of motor binding and unbinding by $k_{on}$ and $k_{off}$, respectively. When defining the first-order rate of motor binding, namely, $k_{on}\rho_{MT}(r)$, we explicitly account for the local microtubule concentration $\rho_{MT}(r)$ extracted from fluorescence images. This is unlike the previous models which imposed specific functional forms on the microtubule distribution (e.g., a constant value [*Lee and Kardar, 2001*; *Sankararaman et al., 2004*] or a power-law decay [*Nédélec et al., 2001*]), rendering them unable to capture the specific features often seen in our measured microtubule profiles, such as the presence of an inflection point (see *Figure 1C* for an example).

From these assumptions, the governing equations for the bound ($m_b$) and free ($m_f$) motor concentrations are shown in *Figure 2C*. They involve binding and unbinding terms, as well as a separate flux divergence term for each population. Solving these equations at steady state, we arrive at an equation for the total local concentration of motors defined as $m_{tot}(r) = m_b(r) + m_f(r)$. The derivation of this result can be found in SI section 'Model formulation'. As seen in the equation for $m_{tot}(r)$ (*Figure 2C*), knowing the microtubule distribution $\rho_{MT}(r)$ along with two effective microscopic parameters, namely, the effective dissociation constant $K_d = k_{off}/k_{on}$ and the length scale $\lambda_0 = D/v$, we can obtain the motor distribution up to a multiplicative constant ($C$ in the equation). Note that in the special case where the motors do not move ($v \rightarrow 0$ or $\lambda_0 \rightarrow \infty$), the exponential term becomes 1 and an equilibrium relation between the motor and microtubule distributions dependent only on $K_d$ is recovered, as we would expect for an equilibrium system.

To test this model, we extract the average radial distributions of microtubule and motor concentrations for each aster. Then, using the microtubule profile as an input, we fit our model to the motor data and infer the effective parameters $K_d$ and $\lambda_0$ (see SI sections 'Extraction of concentration profiles from raw images' and 'Model fitting' for details). A demonstration of this procedure on an example Kif11 aster is shown in *Figure 2D* where a good fit to the average motor data can be observed. As

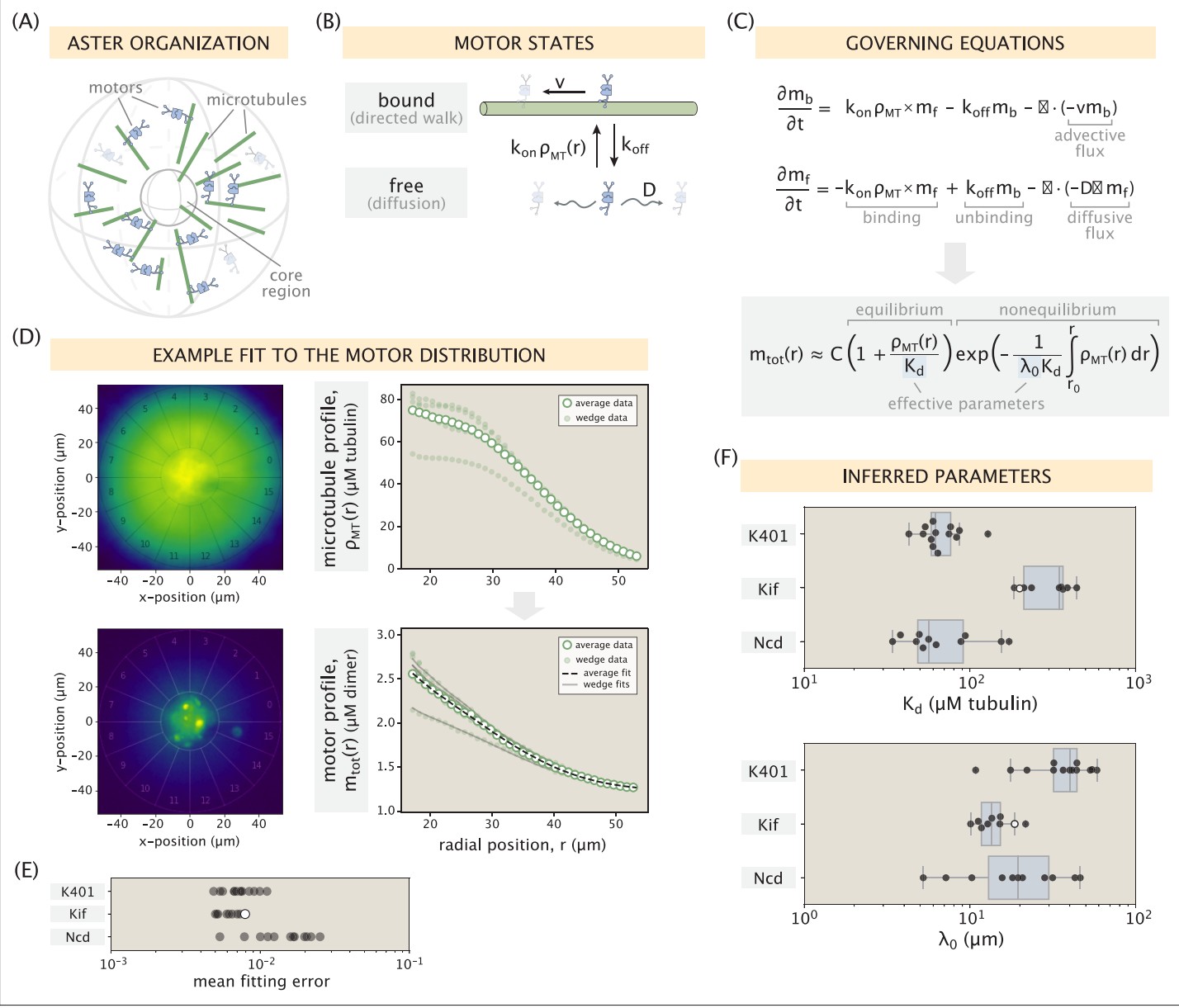

**Figure 2.** Modeling the motor distribution. (**A**) Schematic of the radial microtubule organization in an aster. Modeling applies to locations outside the disordered core region at the aster center. Components of the schematic are not drawn to scale. (**B**) Motor states and transitions between them. (**C**) Governing equations for the bound and free motor populations, along with our solution for the total motor distribution at steady state, expressed via effective parameters $K_d = k_{off}/k_{on}$ and $\lambda_0 = D/v$ (see SI section 'Model formulation' for details). (**D**) Demonstration of the model fitting procedure on an example Kif11 aster. Fits to the average motor profile as well as to 5 out of 16 wedge profiles are shown. The outlier case with a lower concentration corresponds to wedge 13 in the fluorescence images. (**E**) Mean fitting errors for all asters calculated from the fits to the wedge profiles. The error is defined as the ratio of the mean residual to the concentration value at the inner boundary. (**F**) Inferred parameters $K_d$ and $\lambda_0$ grouped by the kind of motor. Box plots indicate the quartiles of the inferred parameter sets. The fitting error and the inferred parameters for the Kif11 aster in panel (**D**) are shown as white dots in panels (**E**) and (**F**).

The online version of this article includes the following figure supplement(s) for figure 2:

**Figure supplement 1.** Collection of all fits to motor profiles.

a validation of our inference method, we additionally extract the radial concentration profiles inside separate wedges of the aster and show that they can be accurately captured by only choosing an appropriate multiplicative constant $C$ for each wedge, while keeping the pair $(K_d, \lambda_0)$ inferred from average profile fixed. Fits to 5 out of 16 different wedge profiles are shown in *Figure 2D* for clarity.

The fitting error for other asters is similarly low (*Figure 2E*, see *Figure 2—figure supplement 1* for the collection of fitted profiles).

Plotting the inferred parameters $K_d$ and $\lambda_0$ from all fits, shown in *Figure 2F*, we find that they are clustered around single values for each motor type and vary between the motors. Based on the single-molecule motor properties in *Table 1* and the reported motor binding rates (*Valentine and Gilbert, 2007*), our expectation was that the $K_d$ values for Kif11 and K401 would have a ratio of $\approx 4.6 : 1$ (see SI section 'Expected ratio of $K_d$ values for K401 and Kif11 motors'), while $K_d$ for Ncd would be the highest due to its non-processivity. The ratio of median inferred $K_d$ values for Kif11 and K401 is $\approx 5.6 : 1$ – close to our expectation. However, the inferred $K_d$ values for Ncd are low and comparable to those for K401.

One possible resolution of this discrepancy comes from the finding of an in vitro study suggesting a substantial increase in the processivity of Ncd motors that act collectively (*Furuta et al., 2013*). Specifically, a pair of Ncd motors coupled through a DNA scaffold was shown to have a processivity reaching 1 μm (or, $\approx 100$ steps) – a value close to that reported for K401 motors. A highly processive movement was similarly observed for clusters of HSET (human kinesin-14) (*Norris et al., 2018*) and plant kinesin-14 motors (*Jonsson et al., 2015*).

This collective effect, likely realized for Ncd tetramers clustered on microtubules in highly concentrated aster structures, is therefore a possible cause for the low inferred values of their effective $K_d$. We also note that while a similar collective effect on processivity was observed for K401 motors (*Furuta et al., 2013*), it is far less dramatic since their single-motor processivity is already about the length of our microtubules, and therefore would have a small effect on the effective $K_d$.

Next, looking at the inference results for the $\lambda_0$ parameter (*Figure 2F*), we can see that Kif11 and Ncd motors have an average $\lambda_0$ value of $\approx 10 - 20$ μm, while the average value for K401 motors is $\approx 40$ μm. From the measured diffusion coefficient of $D \approx 1$ μm²/s for tagged kinesin motors (*Grover et al., 2016*) and the single-molecule motor speeds reported in *Table 1*, our rough estimate for the $\lambda_0$ parameter for Kif11 and Ncd motors was $\approx 10 - 15$ μm, and $\approx 2$ μm for K401. While the inferred values for the two slower motors are well within the order-of-magnitude of our guess, the inferred $\lambda_0$ for the faster K401 motor is much higher than what we anticipated. This suggests a significant reduction in the effective speed. One contributor to this reduction is the stalling of motors upon reaching the microtubule ends. Recall that in our model formulation (*Figure 2B*) we assumed an unobstructed walk for bound motors. Since the median length of microtubules ($\approx 1.6$ μm) is comparable to the processivity of K401 motors ($\approx 1$ μm), stalling events at microtubule ends will be common, leading to a reduction of their effective speed in the bound state by a factor of $\approx 1.5$ (see SI section 'Accounting for finite MT lengths' for details). This correction alone, however, is not sufficient to capture the factor of $\approx 25$ discrepancy between our inference and the estimate of $\lambda_0$. We hypothesize that an additional contribution may come from the jamming of K401 motors in dense aster regions. This is motivated by the experiments which showed that K401 motors would pause when encountering obstructions during their walk (*Ferro et al., 2019*; *Schneider et al., 2015*). In contrast, for motors like Ncd and Kif11 which take fewer steps before unbinding and have a larger effective $K_d$, jamming would have a lesser effect on their effective speed as they would unbind more readily upon encountering an obstacle. Overall, our study shows that the minimal model of motor distributions proposed in *Figure 2* is able to capture the distinctions in aster structure through motor-specific effective parameters, although more work needs to be done to explain the emergence of higher-order effects such as motor clustering and jamming, and their contribution to these effective parameters.

Our model also provides insights on the observation that the distribution of microtubules is generally broader than that of the motors. This feature can be observed by comparing the two example profiles in *Figure 2D*, and it also holds for the profiles extracted from other asters (*Figure 2—figure supplement 1*). In SI section 'Broader spread of the tubulin profile', we use our model to demonstrate this feature in a special analytically tractable case, and discuss its generality across asters in greater detail. We found that the relative width of the motor distribution compared to the microtubule distribution is fairly constant among asters, with their difference being the largest for Ncd motors, consistent with our model predictions. This relationship between the shapes of the distributions may be an important factor in the spatial organization of end-directed motors in the spindle where their localization to the spindle pole is of physiological importance.

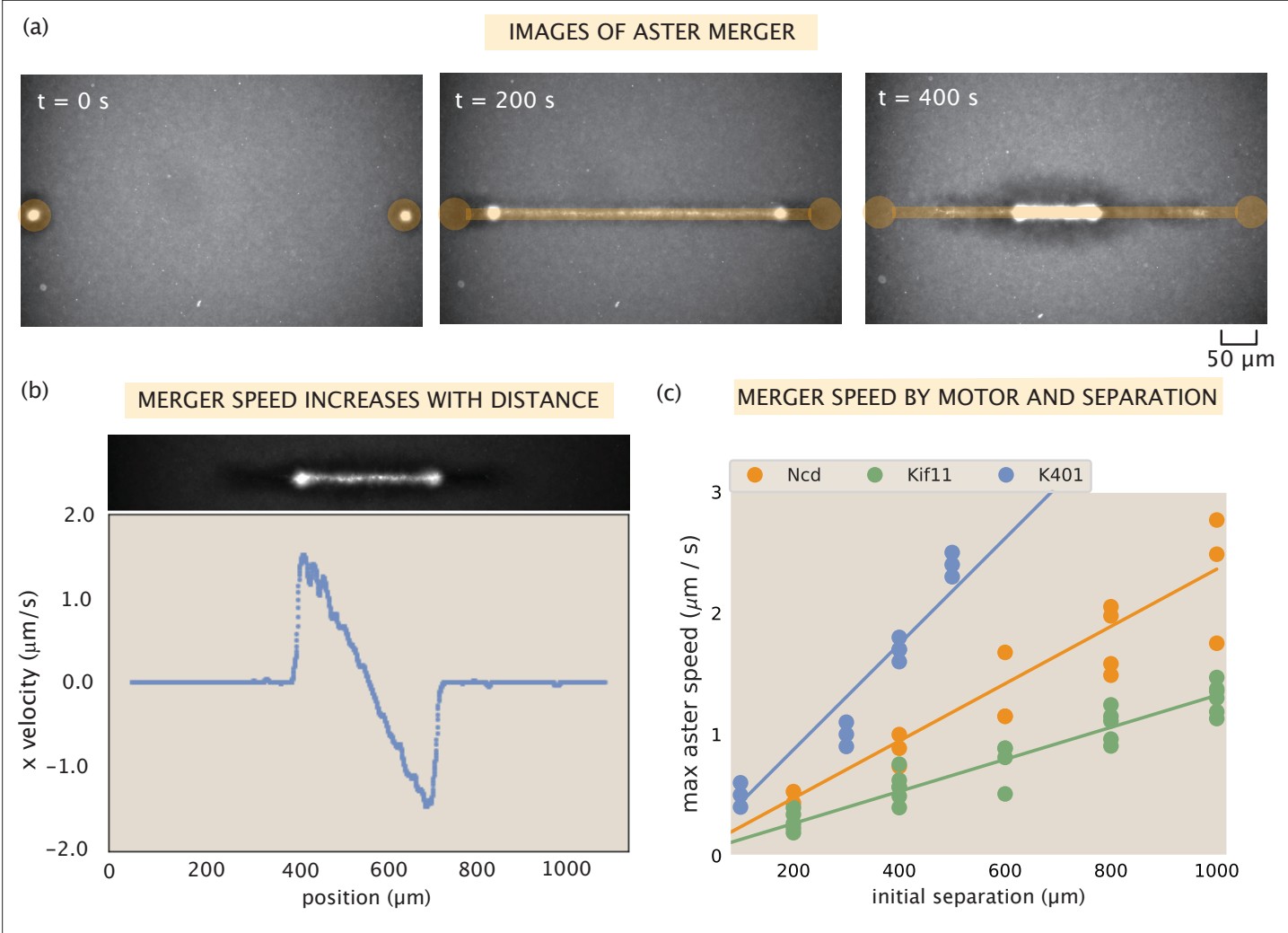

**Figure 3.** Contractile speeds in motor-microtubule networks scale with network size and motor speed. (**a**) Images of microtubule fluorescence during aster merger. Regions of light activation are shown in orange. (**b**) Example profile of speeds in an aster merger as a function of linear distance. Each dot is the mean speed measured at that x-position within the network. (**c**) Maximum merger speed, measured at the ends of the network for each initial separation, and motor. Each dot is a single experiment and the lines are best fits to the data.

The online version of this article includes the following figure supplement(s) for figure 3:

**Figure supplement 1.** Scaling of contractile speeds with network size.

## Contraction rate scales with motor speed

Besides the steady-state structure of motor-microtubule assemblies, it is also of great interest to understand their dynamics. Ross et al. demonstrated the formation of quasi-one-dimensional contractile networks by creating two asters that are initially separated by a given distance, then activating a thin rectangular region between them to form a network connecting the asters that pulls them together (*Ross et al., 2019*). Example images of microtubule fluorescence during one of these experiments are shown in *Figure 3a*. In these experiments, an increase in maximum aster merger speed with greater initial separation was observed. We aimed to confirm this behavior with our various motors and to test the relationship between aster merger speed and single-motor speed.

First, we tested the relationship between distance and speed in our experiments. Using optical flow to measure the contraction speed throughout the network, we observe a linear increase in contractile speed with distance from the center of the network, as shown in *Figure 3b*. This relationship suggests that the contractile network can be thought of as a series of connected contractile units. These findings are in agreement with results from several studies of contractile rates in actomyosin networks,

that suggested telescoping models of contraction, and suggest that this may be a common mechanism across cytosketetal networks (*Thoresen et al., 2011*; *Linsmeier et al., 2016*; *Schuppler et al., 2016*). Independent contraction of each unit would generate the observed linear increase in speed because more contractile units are added with distance from the center of the network.

Next, we investigated how contractile speeds vary by motor in aster merger experiments. We repeated aster merger experiments with each motor and with various initial separations between the asters. *Figure 3c* shows the results, where each point represents the maximum aster speed measured in a single experiment by tracking the aster, and the lines are linear fits to the data for each motor (see SI section 'Merger analysis' for details on measuring the aster speeds). Interestingly, the ratios of the slopes of these lines match the ratios of motor speeds from *Table 1*. For example, the slope of the best fit line for Ncd is $\approx 0.0023\,\mathrm{s}^{-1}$ and the best fit slope for Kif11 is $\approx 0.0013\,\mathrm{s}^{-1}$. The ratio of these (Ncd/Kif11) is $\approx 1.8$, which closely matches the ratio of their single-motor speeds ($\approx 1.6$). Similar calculations can be done with these two motors and K401, with the same result. Thus, we conclude that the rate of contraction in the network is set by the motor speed and the increase of network speed with distance is due to adding more connected contractile units (*Figure 3—figure supplement 1*). It is important to note that we only measure the initial contraction rate in these experiments, over the first $\approx 100$ s. On this time scale, we think of the network as an elastic material, since the time for motor unbinding and rearrangement should be longer than this. The time scale for the optogenetic pair to unbind is about 20 s (*Guntas et al., 2015*); since binding events are independent then the average time for two to unbind is about 400 s. There are likely several motors bound between any two microtubules within the network, thus the relaxation time, or the time to observe the viscous properties of the system, will be the time for multiple motors to unbind or optogenetic links to rearrange. Therefore, to account for the change in contraction rate throughout the process, one would likely need to account for these viscous effects.

## Discussion

In this work, we examined how the properties of kinesin motors determine the mesoscopic properties of the structures they create. The way in which quantities such as motor speed and processivity govern the nature of the resulting motor-microtubule structures has been an open question. Previous attempts have been made to address this question, however most of these are in the context of a single motor. While models could be developed that fit the properties they measure, these models could not be systematically tested since they did not vary the speed or processivity of the motor (*Nédélec et al., 2001*; *Lee and Kardar, 2001*; *Sankararaman et al., 2004*; *Aranson and Tsimring, 2006*). By varying motor speeds, processivities, and directionalities, we were able to quantify and model how these microscopic parameters connect to the properties of mesoscopic structures.

We demonstrate light-controlled aster formation with three different motors. Interestingly, the final aster size from a given illumination region varied depending upon which motor was used. Our leading hypothesis is that the key control variable is the processivity of the motors. Future work needs to be done to understand this effect and build models to explain it. Early work by Surrey et al. found that processivity affected the intensity of aster formation in simulations, which may be related to our observations, but to the best of our knowledge no model of this effect has been developed (*Surrey et al., 2001*). Further, we assess the distribution of motors and microtubules in the asters we form and develop a model of the steady-state aster that predicts the motor distribution given the measured microtubule distribution, with parameters that relate to the motor speed and processivity. Interestingly, the parameters we infer in some cases differ from those we would expect from the single-molecule properties of the motors, indicating that higher-order effects such as cooperativity in collections of motors increasing the processivity of the collective, are playing important roles. In addition, we measure contraction speeds in pseudo-one-dimensional networks and find that the speeds are related to the single-motor speed. In all, this work takes a step toward a mechanistic understanding of motor-microtubule assemblies, translating microscopic properties of individual interactions to the observed properties of the much larger-scale assemblies. This begins to open the door to understanding how different motors and tubulins interact to form cellular scale structures with varying properties, a critical question in evolutionary biology.

## Materials and methods

### Cloning of motor proteins

Human kinesin-5 (Kif11/Eg5) 5–513 was PCR amplified from mCherry-Kinesin11-N-18 plasmid (gift from Michael Davidson, Addgene # 55067). This fragment was previously shown to form functional dimers (*Valentine et al., 2006*). Kinesin 1 1–401 (K401) was PCR amplified from pWC2 plasmid (Addgene # 15960). Ncd 236–701 was PCR amplified from a plasmid gifted by Andrea Serra-Marques.

The optogenetic proteins, iLid, and Micro were PCR amplified from pQE-80L iLid (Addgene # 60408, gift from Brian Kuhlman) and pQE-80L MBP-SspB Micro (Addgene # 60410). mCherry was PCR amplified from mCherry-Kinesin11-N-18 and mVenus was PCR amplified from mVenus plasmid (Addgene # 27793).

Constructs were assembled by Gibson assembly of the desired motor protein, optogenetic protein, and fluorophore in order to make the plasmids listed in *Supplementary file 1*.

### Protein expression and purification

Protein expression and purification was done in SF9 cells. Cells were seeded at a density of 1,000,000 cells per mL in a 15 mL volume and transiently transfected with the desired plasmid using Escort IV transfection reagent, then incubated for 72 hr before purification. Cells were collected for purification by centrifugation at $500 \times g$ for 12 min, and the pellet was resuspended in lysis buffer (200 mM NaCl, 4 mM $MgCl_2$, 0.5 mM EDTA, 1.0 mM EGTA 0.5% Igepal, 7% sucrose by weight, 20 mM imidazole pH 7.5, 10 g/mL aprotinin, 10 g/mL leupeptin, 2 mM ATP, 5 mM DTT, 1 mM PMSF) and incubated on ice for 30 min. The lysate was then clarified by centrifugation at $200,000 \times g$ for 30 min at 4°C. Clarified supernatant was incubated with 40 L anti-FLAG M2 affinity gel (Sigma-Aldrich A2220) for 3 hr at 4°C. To wash out unbound protein, the resin (with bound protein) was collected by centrifugation at $2000 \times g$ for 1 min, the supernatant was removed and the resin was washed with wash buffer (for Ncd and Kif11: 150 mM KCl, 5 mM $MgCl_2$, 1 mM EDTA, 1 mM EGTA, 20 mM imidazole pH 7.5, 10 g/mL aprotinin, 10 g/mL leupeptin, 3 mM DTT, 3 mM ATP; for K401: M2B with 10 g/mL aprotinin, 10 g/mL leupeptin, 3 mM DTT, 3 mM ATP). This was repeated two more times with decreasing ATP concentration (0.3 and 0.03 mM ATP) for a total of three washes. After the third wash, about 100 L supernatant was left and the bound protein was eluted by incubation with 10 L FLAG peptide (Sigma-Aldrich F3290) at 4°C for 3 hr. The resin was then spun down by centrifugation at $2000 \times g$ for 1 min and the supernatant containing the purified protein was collected. Purified protein was then concentrated to a volume of 10–20 L by centrifugation in mini spin filters (Millipore 50 kDa molecular weight cut-off). Protein was kept at 4°C and used the same day as purification or stored in 50% glycerol at –20°C for longer storage. Protein concentration was determined with QuBit Protein Assay Kit (Thermo Fisher Q33212).

### Microtubule polymerization

Microtubules were polymerized as reported previously (*Ross et al., 2019*) and originally from the Mitchison lab website (*Georgoulia, 2012*). In brief, 75 µM unlabeled tubulin (Cytoskeleton) and 5 µM tubulin-Alexa Fluor 647 (Cytoskeleton) were combined with 1 mM DTT and 0.6 mM GMP-CPP in M2B buffer and incubated spun at $300,000 \times g$ to remove aggregates, then the supernatant was incubated at 37°C for 1 hr to form GMP-CPP stabilized microtubules.

### Microtubule length

The GMP-CPP stabilized microtubules were imaged with TIRF microscopy to determine their length. A flow chamber was made using a KOH cleaned slide, KOH cleaned coverslip (optionally coated with polyacrylamide), and parafilm cut into chambers. The flow cell was incubated with poly-L-lysine for 10 min, washed with M2B, then microtubules were flown in. The chamber was sealed with Picodent and imaged with TIRF microscopy.

Microtubules were segmented using home-written Python code and histogrammed to determine the distribution of microtubule lengths (*Appendix 1—figure 1*).

## Sample chamber preparation

Slides and coverslips were cleaned with Helmanex, ethanol, and KOH, silanized, and coated with polyacrylamide as in *Ross et al., 2019*. Just before use, slides and coverslips were rinsed with MilliQ water and dried with compressed air. Flow chambers (3 mm wide) were cut out of Parafilm M and melted using a hotplate at 65°C to seal the slide and coverglass together, forming chambers that are $\approx 70 - 100$ μm in height and contain ≈7 μL.

## Reaction mixture preparation

The reaction mixture consisted of kinesin motors ($\sim 250\,\mathrm{nM}$), microtubules (~1 μM tubulin), and energy mix that contained ATP, an ATP recycling system, a system to reduce photobleaching, F-127 pluronic to reduce interactions with the glass surfaces, and glycerol (*Ross et al., 2019*). To prevent pre-activation of the optogenetic proteins and photobleaching of the fluorophores, the motors and microtubules were always handled in a dark room where wavelengths of light below 520 nm were blocked with a filter or a red light was used to illuminate. The reaction mixture was prepared right before loading into the flow cell and then sealed with Picodent Speed.

## Microscope instrumentation

We performed the experiments with an automated widefield epifluorescence microscope (Nikon TE2000). We custom modified the scope to provide two additional modes of imaging: epi-illuminated pattern projection and LED gated transmitted light. We imaged light patterns from a programmable DLP chip (EKB TEchnologies DLP LightCrafter E4500 MKIITM Fiber Couple) onto the sample through a user-modified epi-illumination attachment (Nikon T-FL). The DLP chip was illuminated by a fiber coupled 470 nm LED (ThorLabs M470L3). The epi-illumination attachment had two light-path entry ports, one for the projected pattern light path and the other for a standard widefield epi-fluorescence light path. The two light paths were overlapped with a dichroic mirror (Semrock BLP01-488R-25). The magnification of the epi-illuminating system was designed so that the imaging sensor of the camera (FliR BFLY-U3-23S6M-C) was fully illuminated when the entire DLP chip was on. Experiments were run with Micro-Manager (*Edelstein et al., 2010*), running custom scripts to controlled pattern projection and stage movement.

## Activation and imaging protocol

For the experiments in which we make asters with excitation disks of different sizes, we use five positions within the same flow cell simultaneously in order to control for variation within flow cells and over time. Each position is illuminated with a different sized excitation region: 50, 100, 200, 400, or 600 μm diameter cylinder. Each position was illuminated with the activation light for $\sim 50 - 200$ ms and both the microtubules (Cy5 labeled) and motors (mVenus labeled) were imaged at ×10 magnification every 15 s. After an hour of activation, a z-stack of the microtubule and motor fluorescence throughout the depth of the flow chamber was taken at 5 μm increments in each position. Typically, one experiment was run per flow chamber. We placed the time limitations on the sample viewing to minimize effects related to cumulative photobleaching, ATP depletion, and global activity of the light-dimerizable proteins. After several hours, inactivated dark regions of the sample begin to show bundling of microtubules.

For the aster merger experiments, two 50 μm disks are illuminated at different distances apart. Again, five positions within the same flow cell are chosen, and the separation between the two disks varies for each position: 200, 400, 600, 800, or 1000 μm apart. For K401 and Ncd experiments, the disks are illuminated for 30 frames, for Kif11 experiments, the disks are illuminated for 60 frames (frames are every 15 s). Then, a bar ≈5 μm wide connecting the disks is illuminated to merge the asters.

## Gliding assay

Motor speeds were determined by gliding assay. Glass slides and coverslips were Helmanex, ethanol, and KOH cleaned. Flow cells with ≈10 L volume were created with double sided sticky tape, and rinsed with M2B buffer. Then, anti-GFP antibody was applied and incubated for 10 min. The flow cell was then rinsed with M2B and then mVenus labeled motor proteins (at $\sim$ 5 nM in M2B) were flowed in and incubated for 10 min. The flow cell was rinsed with M2B to remove unbound motors and microtubules (in M2B with 3 mM ATP and 1 mM DTT) were flowed in. Microtubules were then imaged using

total internal reflection fluorescence (TIRF) microscopy at a rate of one frame per second. Individual microtubules were tracked using custom-written Python code to determine their speed. The mean microtubule speed (excluding those that were not moving) was determined as the motor speed. *Appendix 1—figure 2* shows the histogram of speeds obtained for K401 motors purified from SF9 cells.

## Forming a single aster

The experiments for this study were performed in a regime in which we obtained a single aster, in order to measure properties of the structure and compare between various motors. However, by varying concentrations of components within the system such as motor and microtubule concentration, it is possible to obtain different results. Some examples include mini asters everywhere in the sample before activation, a few asters within the activation region, or many small asters within the activation region. Example images of these cases are shown in *Figure 1—figure supplement 1*. The various possible resulting structures, and the perturbations to the system to obtain them, warrant further study in the future.

## Disordered aster core

We observe that the asters we create have centers that are very dense with motors and microtubules. By fluorescence microscopy, we do not observe organized aster arms in this region and hypothesized that the microtubules are disordered in this region. To assess the extent of microtubule organization in our asters, we imaged asters with a polarized light microscope (Pol-Scope). This microscope utilizes polarized white light to image birefringent substances. Microtubules are birefringent due to their aspect ratio; they interact differently with light polarized parallel to their long axes compared to light polarized perpendicular to their long axes. Thus, the Pol-Scope allows determination of the alignment of microtubules, but not their plus/minus end polarity (*Oldenbourg and Mei, 1995*). When imaged with a Pol-Scope, the arms of our asters are bright, indicating high alignment, and their azimuthal angle confirms that they are radially symmetric around the center (*Appendix 1—figure 3*). The center of the aster is dark, which we interpret to mean that this region is disordered. It is possible that the microtubules in the center could be aligned pointing in the z-direction, which could also result in the center being dark. A disordered center may be a result from steric hindrance due to a high density of microtubules in that region, which prevents the motors from aligning the microtubules. Due to the disorder in the aster center, we exclude this region from our theoretical analysis.

## Steady-state aster

When we measure aster size and the distributions of motors and microtubules, we do so once the aster has reached a dynamic steady state. To assess this notion, we performed FRAP experiments. We photobleached the microtubules in a fully formed aster in a grid pattern and took images of the recovery, shown in *Appendix 1—figure 4*. From these images, it is clear that there is little to no net radial movement of the microtubules, and slow angular motion. Therefore, at this point, the aster is no longer contracting, but the microtubules are still dynamic. We attempted similar experiments with the motors, but they recover very quickly.

## Aster size

We formed asters of various sizes using cylindrical illumination regions, ranging in diameter from 50 to 600 μm. *Figure 1—figure supplement 2* shows representative images of the microtubule fluorescence of asters formed with each motor and each excitation diameter. The yellow circles are by-hand determination of the outer boundary of the aster. It can be seen in these images that for the smallest asters, the aster size can be larger than the illumination region. We believe this is due to diffusion of dimerized motors outside of the activation region, where they can bind microtubules in the background region and incorporate them into the asters. For larger excitation diameters, the microtubules in the activated region get concentrated in the aster, leaving a region depleted of microtubules between the aster and the background region. In order to measure the size of asters in a more systematic way, we used the measured microtubule distribution, as shown in *Figure 1C*. Outside of the central core region of the aster, microtubule fluorescence decreases monotonically before rising again to the background level outside of the activation region. We chose to define the outer radius of

the aster as the radius at which the median microtubule fluorescence is twice the background fluorescence. This metric agrees well with a visual inspection of the asters.

## Acknowledgements

We are grateful to Dan Needleman, Madhav Mani, Peter Foster, and Ana Duarte for fruitful discussions. We acknowledge support from the NIH through grant 1R35 GM118043-01; the John Templeton Foundation as part of the Boundaries of Life Initiative through grants 51250 and 60973; the Foundational Questions Institute and Fetzer Franklin Fund through FQXi 1816.

## Additional information

### Funding

| Funder | Grant reference number | Author |
|---|---|---|
| Foundational Questions Institute | FQXi 1816 | Rachel A Banks<br>Matt Thomson<br>Rob Phillips<br>Soichi Hirokawa<br>Heun Jin Lee<br>Vahe Galstyan |
| John Templeton Foundation | 51250 | Rachel A Banks<br>Vahe Galstyan<br>Heun Jin Lee<br>Soichi Hirokawa<br>Matt Thomson<br>Rob Phillips |
| National Institutes of Health | 1R35 GM118043-01 | Rachel A Banks<br>Vahe Galstyan<br>Heun Jin Lee<br>Soichi Hirokawa<br>Rob Phillips |
| John Templeton Foundation | 60973 | Rachel A Banks<br>Vahe Galstyan<br>Heun Jin Lee<br>Soichi Hirokawa<br>Matt Thomson<br>Rob Phillips |

The funders had no role in study design, data collection and interpretation, or the decision to submit the work for publication.

### Author contributions

Rachel A Banks, Conceptualization, Data curation, Software, Formal analysis, Validation, Investigation, Visualization, Methodology, Writing - original draft, Project administration, Writing – review and editing; Vahe Galstyan, Software, Formal analysis, Visualization, Methodology, Writing - original draft, Writing – review and editing; Heun Jin Lee, Conceptualization, Resources, Data curation, Supervision, Investigation, Methodology, Project administration, Writing – review and editing; Soichi Hirokawa, Software, Formal analysis, Methodology, Writing – review and editing; Athena Ierokomos, Resources, Methodology, Writing – review and editing; Tyler D Ross, Zev Bryant, Conceptualization, Resources, Methodology, Writing – review and editing; Matt Thomson, Resources, Funding acquisition, Writing – review and editing; Rob Phillips, Conceptualization, Supervision, Funding acquisition, Project administration, Writing – review and editing

### Author ORCIDs

Rachel A Banks  http://orcid.org/0000-0003-2028-2925
Vahe Galstyan  http://orcid.org/0000-0001-7073-9175
Soichi Hirokawa  http://orcid.org/0000-0001-5584-2676
Rob Phillips  http://orcid.org/0000-0003-3082-2809

**Decision letter and Author response**

Decision letter https://doi.org/10.7554/eLife.79402.sa1
Author response https://doi.org/10.7554/eLife.79402.sa2

## Additional files

### Supplementary files

• Supplementary file 1. Plasmids generated for and used in this study.

• Supplementary file 2. Processivities of motors, decay length scales of motor profiles, and corresponding ratios of these two length scales. Step size of $\approx 10$ nm corresponding to the length of a tubulin dimer was used for estimating the motor processivities in μm units. For Ncd motors, the upper limit in processivity corresponds to that of oligomeric motor assemblies. Estimates for the decay length scales $\lambda$ were made based on the motor profiles in *Figure 2—figure supplement 1*.

• MDAR checklist

### Data availability

All data associated with this study are stored on the CaltechData archive at https://doi.org/10.22002/D1.2152.

The following previously published dataset was used:

| Author(s) | Year | Dataset title | Dataset URL | Database and Identifier |
|---|---|---|---|---|
| Banks RA, Galstyan V, Lee HJ, Hirokawa S, Ierokomos A, Ross TD, Bryant Z, Thomson M, Phillips R | 2021 | Images for Motor processivity and speed determine structure and dynamics of motor-microtubule assemblies | https://doi.org/10.22002/D1.2152 | CaltechDATA, 10.22002/D1.2152 |

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

## Appendix 1

### Motor distributions
Model formulation

To predict the spatial distribution of motors in aster structures, we model the dynamic steady state of free (*f*) and bound (*b*) motor concentrations via

$$\frac{\partial m_b}{\partial t} = k_{on}\rho_{MT}(\mathbf{r})m_f(\mathbf{r}) - k_{off}m_b(\mathbf{r}) - \nabla \cdot \mathbf{J}_v = 0, \tag{1}$$

$$\frac{\partial m_f}{\partial t} = -k_{on}\rho_{MT}(\mathbf{r})m_f(\mathbf{r}) + k_{off}m_b(\mathbf{r}) - \nabla \cdot \mathbf{J}_D = 0. \tag{2}$$

Here, $k_{on}$ and $k_{off}$ are the motor binding and unbinding rates, respectively, $\rho_{MT}(\mathbf{r})$ is the spatially varying microtubule concentration at steady state (measured as µM tubulin), $\mathbf{J}_v$ is the advective flux of bound motors, and $\mathbf{J}_D$ is the diffusive flux of free motors. Our modeling approach is similar to that used by *Nédélec et al., 2001*, with the main difference being in the handling of $\rho_{MT}(\mathbf{r})$. Namely, they imposed a particular functional form on this distribution ($\rho_{MT}(\mathbf{r}) = 1/|\mathbf{r}|^{d-1}$ with $d$ as the spatial dimension) based on an idealized representation of microtubule organization in an aster, whereas in our treatment $\rho_{MT}(\mathbf{r})$ stands for the experimentally measured microtubule profiles which cannot be captured through an analogous idealization.

If the free motors have a diffusion coefficient $D$, then, in the radially symmetric setting considered in our modeling, the diffusive flux will be given by

$$\mathbf{J}_D(r) = -D\nabla m_f(\mathbf{r}) = -Dm_f'(r)\hat{\mathbf{r}}, \tag{3}$$

where $\hat{\mathbf{r}}$ is an outward-pointing unit radial vector. And if $v$ is the advection speed of bound motors, then the advective motor flux on radially organized microtubules will be

$$\mathbf{J}_v(r) = -vm_b(r)\hat{\mathbf{r}}. \tag{4}$$

Here, we are implicitly assuming that motors constantly walk when bound, ignoring the fact that they can stall upon reaching a microtubule end. We discuss the impact of this effect later in 'Accounting for finite MT lengths'.

At steady state, the net flux of motors at any radial distance $r$ should be zero ($\mathbf{J}_D(r) + \mathbf{J}_v(r) = 0$), which implies a general relation between the profiles of free and bound motors, namely,

$$\begin{aligned} 0 &= -Dm_f'(r)\hat{\mathbf{r}} - vmb(r)\hat{\mathbf{r}} \Rightarrow \\ m_b(r) &= -\underbrace{\left(\frac{D}{v}\right)}_{\lambda_0} m_f'(r) \end{aligned} \quad . \tag{5}$$

Above we introduced $\lambda_0$ as a length scale parameter that can be interpreted as the distance which is traveled by free and bound motors at similar time scales, that is, diffusion time scale ($\lambda_0^2/D$)=advection time scale ($\lambda_0/v$). Note also that the − sign at the right-hand side indicates that the free motor population should necessarily have a decaying radial profile ($m_f'(r) < 0$) which is intuitive since at steady state the outward diffusion needs to counteract the inward advection.

To make further analytical progress, we will assume that motor binding and unbinding events are locally equilibrated (*Aranson and Tsimring, 2006*). This assumption is valid if motor transport is sufficiently slow compared with binding/unbinding reactions. We will justify this quasi-equilibrium condition for the motors used in our study at the end of the section. It follows from this condition that

$$\begin{aligned} k_{off}m_b(r) &\approx k_{on}\rho_{MT}(r)m_f(r) \Rightarrow \\ m_b(r) &\approx \frac{\rho_{MT}(r)}{K_d} m_f(r) \end{aligned} \quad , \tag{6}$$

where $K_d = k_{off}/k_{on}$ is the dissociation constant. Since the experimental readout reflects the *total* motor concentration ($m_{tot} = m_f + m_b$), we use our results (*Equation 5* and *Equation 6*) to link $m_{tot}(r)$ with the microtubule profile $\rho_{MT}(r)$. Specifically, using *Equation 6* we find

$$
\begin{aligned}
m_{\text{tot}}(r) &= m_{\text{b}}(r) + m_{\text{f}}(r) \\
&= \left(1 + \frac{\rho_{\text{MT}}(r)}{K_{\text{d}}}\right) m_{\text{f}}(r) \Rightarrow
\end{aligned}
\tag{7}
$$

$$
m_{\text{f}}(r) = \frac{K_{\text{d}}}{K_{\text{d}} + \rho_{\text{MT}}(r)} m_{\text{tot}}(r),
\tag{8}
$$

$$
m_{\text{b}}(r) = \frac{\rho_{\text{MT}}(r)}{K_{\text{d}} + \rho_{\text{MT}}(r)} m_{\text{tot}}(r).
\tag{9}
$$

Next, substituting the above expressions for $m_{\text{f}}$ and $m_{\text{b}}$ into *Equation 5* and simplifying, we relate the motor and microtubule profiles, namely,

$$
\begin{aligned}
\frac{\rho_{\text{MT}}(r)}{K_{\text{d}}+\rho_{\text{MT}}(r)} m_{\text{tot}}(r) &= -\lambda_0 \underbrace{\left(\frac{K_{\text{d}}}{K_{\text{d}} + \rho_{\text{MT}}(r)} m_{\text{tot}}'(r) - \frac{K_{\text{d}}\rho_{\text{MT}}'(r)}{(K_{\text{d}} + \rho_{\text{MT}}(r))^2} m_{\text{tot}}(r)\right)}_{m_{\text{f}}'(r)} \Rightarrow \\
\frac{m_{\text{tot}}'(r)}{m_{\text{tot}}(r)} &= -\frac{\rho_{\text{MT}}(r)}{K_{\text{d}}\lambda_0} + \frac{\rho_{\text{MT}}'(r)}{K_{\text{d}} + \rho_{\text{MT}}(r)} \Rightarrow \\
\left(\ln m_{\text{tot}}(r)\right)' &= -\frac{\rho_{\text{MT}}(r)}{K_{\text{d}}\lambda_0} + \left(\ln K_{\text{d}} + \rho_{\text{MT}}(r)\right)' \Rightarrow \\
\ln m_{\text{tot}}(r) &= -\frac{R_{\text{MT}}(r)}{K_{\text{d}}\lambda_0} + \ln(K_{\text{d}} + \rho_{\text{MT}}(r)) + C_1 \Rightarrow \\
m_{\text{tot}}(r) &= C\left(1 + \frac{\rho_{\text{MT}}(r)}{K_{\text{d}}}\right)\exp\left(-\frac{R_{\text{MT}}(r)}{K_{\text{d}}\lambda_0}\right),
\end{aligned}
\tag{10}
$$

where $R_{\text{MT}}(r) = \int \rho_{\text{MT}}(r)\,\mathrm{d}r$ is the integrated microtubule concentration, and $C = K_{\text{d}}\,e^{C_1}$ is a positive constant. The presence of the multiplicative constant $C$ is a consequence of the fact that the two equations used for deriving our result (*Equation 5* and *Equation 6*) specify the *ratios* of motor populations. Therefore, the result in *Equation 10* predicts the relative level of the total motor concentration, given the two effective model parameters ($K_{\text{d}}$ and $\lambda_0$), which we infer in our fitting procedure.

Note that the two variable factors on the right-hand side of *Equation 10* have qualitatively different structures. The first one is local and depends only on the dissociation constant (an equilibrium parameter), while the second term involves an integrated (hence, non-local) microtubule density term and $\lambda_0 = D/v$ which depends on the advection speed $v$ (a non-equilibrium parameter). As anticipated, in the limit of vanishingly slow advection ($v \to 0$ or, $\lambda_0 \to \infty$) the second factor becomes 1 and an equilibrium result is recovered.

## Connections to related works

Before proceeding further into our analysis, we briefly compare the expression for the motor distribution (*Equation 10*) with analogous results in the literature. Specifically, *Nédélec et al., 2001*, studied quasi-two-dimensional asters and in their modeling treated microtubules as very long filaments, all converging at the aster center. This setting implied $\sim 1/r$ scaling of the microtubule concentration. With this scaling, the integrated microtubule concentration in our framework becomes $R_{\text{MT}}(r) = \int \frac{\alpha}{r}\,\mathrm{d}r = \alpha \ln r$, where $\alpha$ is a constant. Substituting this form into the exponential term in *Equation 10*, we find $\exp\{-(K_{\text{d}}\lambda_0)^{-1} R_{\text{MT}}(r)\} = \exp\{-\alpha(K_{\text{d}}\lambda_0)^{-1}\ln r\} \sim 1/r^{\beta}$, where $\beta = \alpha(K_{\text{d}}\lambda_0)^{-1}$. It then follows from *Equation 10* and the scaling $\rho_{\text{MT}}(r) \sim 1/r$ that the motor concentration is a sum of two decaying power laws (namely, $\sim 1/r^{\beta}$ and $\sim 1/r^{\beta+1}$) the result obtained by *Nédélec et al., 2001*. A more detailed calculation can be done to demonstrate that the exponent $\beta$ matches exactly with the result derived in the earlier work, but for the purposes of our study we do not elaborate further on this comparison. We note that the experimentally measured microtubule profiles in asters (e.g., *Figure 2B* or *Figure 3d*) often have an inflection point and cannot be fitted to decaying power-law functions (e.g., $1/r^2$ for 3D asters), which is why the idealized setting considered by *Nédélec et al., 2001*, cannot be applied to our system.

Another set of works (*Lee and Kardar, 2001*; *Sankararaman et al., 2004*) also studied motor distributions in asters, but this time under the assumption of a uniform microtubule concentration ($\rho_{\text{MT}}(r) \sim$ constant). In such a setting, our framework predicts an exponentially decaying motor profile, because $R_{\text{MT}}(r) = \int \rho_{\text{MT}}(r)\,\mathrm{d}r \sim \rho_{\text{MT}} r$ and thus, $m_{\text{tot}}(r) \sim e^{-\rho_{\text{MT}} r/K_{\text{d}}\lambda_0}$. An exponential decay was also the prediction of *Lee and Kardar, 2001*, although in their treatment all motors were assumed to be in the bound state. The two distinct motor states were considered in the work by *Sankararaman*

*et al., 2004*, who predicted an exponential decay of motor concentration modulated by a power-law tail. One can show, however, that when the decay length scale of motor concentration greatly exceeds the motor processivity (as in the case of asters which we generated), the prediction of *Sankararaman et al., 2004*, also reduces into a pure exponential decay, matching the prediction of our model. But since the assumption of a uniform microtubule profile is clearly violated in our system, these predictions are not applicable for us.

## Validity of the quasi-equilibrium assumption

Earlier in the section, we assumed that motor binding and unbinding reactions were locally equilibrated, from which *Equation 6* followed. Looking at the governing equation of bound motor dynamics (*Equation 1*), we can see that this assumption will hold true if $k_{\text{off}} m_{\text{b}}(r) \gg |\nabla \cdot \mathbf{J}_{\text{v}}|$. Substituting the expression of advective flux (*Equation 4*) and recalling that in three dimensions the divergence of a radial vector $\mathbf{A} = A\hat{\mathbf{r}}$ takes the form $r^{-2}\partial_{\text{r}}(r^2 A)$, we rewrite the quasi-equilibrium condition as

$$k_{\text{off}} m_{\text{b}}(r) \gg |\nabla \cdot (-v m_{\text{b}}(r)\hat{\mathbf{r}})| \Rightarrow \tag{11}$$

$$\frac{k_{\text{off}}}{v} m_{\text{b}}(r) \gg \left| m_{\text{b}}'(r) + \frac{2}{r} m_{\text{b}}(r) \right|. \tag{12}$$

Now, many of the motor profiles can be approximated reasonably well by an exponentially decaying function (see *Figure 2—figure supplement 1* for a collection of experimental profiles). This suggests an empirical functional form $m_{\text{b}}(r) \sim e^{-r/\lambda}$ for the concentration of bound motors, where $\lambda$ is the decay length scale (note that the constant saturation level contributes to the *free* motor population). This functional form implies that $m_{\text{b}}'(r) \approx -m_{\text{b}}(r)/\lambda$, which, upon substituting into *Equation 11*, leads to

$$\underbrace{\frac{k_{\text{off}}}{v}}_{\lambda_{\text{v}}^{-1}} m_{\text{b}}(r) \gg \left| -\frac{1}{\lambda} m_{\text{b}}(r) + \frac{2}{r} m_{\text{b}}(r) \right| \Rightarrow \tag{13}$$

$$\frac{\lambda}{\lambda_{\text{v}}} \gg \left| \frac{2\lambda}{r} - 1 \right|, \tag{14}$$

where $\lambda_{\text{v}} = v/k_{\text{off}}$ is introduced as the motor processivity (distance traveled before unbinding). The processivities ($\lambda_v$) of the three different kinesins used in our study together with the observed ranges of decay length scales ($\lambda$) of corresponding motor profiles are listed in *Supplementary file 2*. As can be seen, in all cases the ratio $\lambda/\lambda_v$ is much greater than one, verifying the intuitive expectation that the length scales of aster structures are much greater than the single run lengths of motors.

It is obvious from the presence of the $r^{-1}$ term on the right-hand side of *Equation 13* that the condition can only be satisfied past a certain radius, since $r^{-1}$ becomes very large when $r$ approaches zero. This threshold radius ($r^*$) is set by $r^* \sim 2\lambda_v$, where the two sides of *Equation 13* become comparable. The threshold radial distance that we choose to isolate the core is at least $5 - 10$ µm for the asters of our study (see the lower $x$-limits in the profiles of *Figure 2—figure supplement 1*) which exceed $r^*$ at least a few times. This suggests that *Equation 13* is valid, justifying our use of the quasi-equilibrium assumption for modeling the motor distribution.

## Extraction of concentration profiles from raw images

In this section, we describe our approach for extracting the radial profiles of motor and microtubule concentrations from raw fluorescence images.

### Fluorescence normalization and calibration

When taking images with a microscope, several sources contribute to the detected pixel intensities: the camera offset, autofluorescence from the energy mix, and fluorescence coming from the tagged proteins (tubulin or motors). In addition, due to the uneven illumination of the field of view, the same protein concentration may correspond to different intensities in the raw image.

We begin the processing of raw images by first correcting for the uneven illumination. For microtubule images, we use the first movie frames as references with a uniform tubulin concentration

in order to obtain an intensity normalization matrix. Each pixel intensity of the final image frame is then rescaled by the corresponding normalization factor.

Although the motor concentration is also initially uniform, the light activation region in the first frame appears photobleached, making it unsuitable for the construction of a normalization matrix. Instead, we obtain this matrix from the final frame, after masking out the neighboring region of the aster, outside of which the nonuniformity of the fluorescence serves as a proxy for uneven illumination. Intensity normalization factors inside the masked out circular region are obtained through a biquadratic interpolation scheme. The steps leading to a normalized motor image are depicted in *Appendix 1—figure 5a*.

After fluorescence normalization, we convert intensities into units of protein concentration using calibration factors estimated from images of samples with known protein contents. For K401 and Kif11 motors, we use the conversion 1000 intensity units → 815 nM motor dimer. For Ncd dimers, which have fluorescent tags on both iLid and Micro units, we use the 1000 intensity units → 407 nM conversion. In all three cases, 200 ms exposure time is used in the imaging. For tubulin, we make a rough estimate that after spinning the energy mix with tubulin, around 1 μM of tubulin remain, all of which polymerize into GMP-CPP stabilized microtubules. This leads to the calibration of 360 intensity units → 1 μM tubulin (100 ms exposure time).

## Aster center identification

In the next step of the profile extraction pipeline, we crop out the aster region from the normalized image and identify the aster center in an automated fashion. In particular, we divide the aster into 16 equal wedges, calculate the radial profile of motors within each wedge, and define the aster center as the position that yields the minimum variability between the motor profiles extracted from the different wedges. Having identified the center, a mean radial profile for the aster is defined as the average of the 16 wedge profiles (*Appendix 1—figure 5b*).

## Inner and outer boundary determination

Since our modeling framework applies to regions of the aster where the microtubules are ordered, we consider the concentration profiles in a limited radial range for the model fitting procedure. As we do not have a Pol-Scope image for every aster to precisely identify the disordered core region, we prescribe a lower threshold on the radial range by identifying the position of the fastest intensity drop and adding to it a buffer interval (equal to 15% of the outer radius) to ensure that the region of transitioning from the disordered core into the ordered aster arms is not included (*Appendix 1—figure 5c*, top panel). As for the outer boundary, we set it as the radial position where the tubulin concentration exceeds its background value by a factor of two (*Appendix 1—figure 5c*, bottom panel).

## Model fitting

Here, we provide the details of fitting the expression we derived for the motor distribution (*Equation 10*) to the profiles extracted from aster images. Since smaller asters are typically irregular and hence, do not meet the polar organization and radial symmetry assumptions of the model, we constrain the fitting procedure to larger asters formed in experiments with a minimum light illumination disk diameter of 200 μm.

The different aspects of the fitting procedure are demonstrated in *Appendix 1—figure 6*. Extracting the average tubulin and motor profiles, we fit our model to the motor profile and obtain the optimal values of the effective parameters $K_d$ and $\lambda_0$. With the exception of a few cases, the optimal pair $(K_d, \lambda_0)$ corresponds to a distinct peak in the residual landscape (*Appendix 1—figure 6c*, note the logarithmic scale of the colorbar), suggesting that the parameters are well defined. The fit to the motor data for the example aster is shown in *Appendix 1—figure 6d*.

As stated in the main text, we then use the data from the separate aster wedges to assess the quality of fit for each aster. Specifically, keeping the $(K_d, \lambda_0)$ pair inferred from the average profile fixed, we fit the 16 separate wedge profiles by optimizing over the scaling coefficient $C$ for each of the wedges, and use the model residuals to assess the fit quality. In the set shown in *Appendix 1—figure 6e*, with the exception of wedge 9 which contains an aggregate near the core radius, fits to all other wedge profiles are of good quality, translating into a low fitting error reported in *Figure 2E* of the main text.

Repeating this procedure for all other asters, we obtain the best fits to their motor profiles and the corresponding values of the optimal $(K_d, \lambda_0)$ pairs. The collection of all average profiles, along with the best model fits and inferred parameters are shown in *Figure 2—figure supplement 1*.

## Expected ratio of $K_d$ values for K401 and Kif11 motors

Here, we show the steps in estimating the expected ratio of $K_d$ values for the motors K401 and Kif11. Recall the definition $K_d = k_{off}/k_{on}$. Knowing the motor processivities and speeds from *Table 1*, we calculate the off-rates as $k_{off} = \text{speed/processivity}$. This yields a ratio $k_{off}^{Kif11}/k_{off}^{K401} \approx 1.15$. Then, using the reported on-rates in *Valentine and Gilbert, 2007*, we find the ratio of on-rates to be $k_{on}^{Kif11}/k_{on}^{K401} \approx 0.25$. Taken together, these two results lead to our estimate for the ratio of $K_d$ values reported in the main text, namely, $K_d^{Kif11}/K_d^{K401} = (k_{off}^{Kif11}/k_{off}^{K401})/(k_{on}^{Kif11}/k_{on}^{K401}) \approx 4.6$.

## Accounting for finite MT lengths

Analysis of purified microtubule images shows that the median length of microtubules is $\approx 1.6\ \mu m$ (*Appendix 1—figure 1*). Taking the size of a tubulin dimer to be 8 nm, this length translates into the distance traveled in $\approx 200$ motor steps, which is comparable to the processivity of K401 motors reported in *Table 1* of the main text. Since motors stall when reaching microtubule ends, their effective advection speed will get reduced. Here, we account for this reduction and estimate its magnitude for the different motors used in our study.

Consider the schematic in *Appendix 1—figure 3* where a motor is shown advecting on a microtubule with length $L$. If the distance $x$ between the motor and microtubule end at the moment of binding is less than the motor processivity $\lambda_v$, then the motor will reach the end and stall for a time period $\tau = 1/k_{off}^{end}$ before unbinding. On the other hand, if $x$ is greater than $\lambda_v$, the motor will not stall while bound to the microtubule and hence, its effective speed will not be reduced.

Assuming that the location of initial binding is uniformly distributed in the $[0, L]$ interval (hence, the chances of binding between $x$ and $x + dx$ is $dx/L$), we can calculate the effective speeds in the above two cases as

$$
\begin{aligned}
v_{eff}(L < \lambda_v) \quad &= L^{-1} \in t_0^L \frac{x}{x/v + \tau}\, dx, \\
&= v\left(1 - \frac{v\tau}{L} \ln\left(1 + \frac{L}{v\tau}\right)\right)
\end{aligned}
\tag{15}
$$

$$
\begin{aligned}
v_{eff}(L > \lambda_v) \quad &= \underbrace{L^{-1} \int_0^{\lambda_v} \frac{x}{x/v + \tau}\, dx}_{\text{initial position} < \lambda_v} + \underbrace{L^{-1} \int_{\lambda_v}^L v\, dx}_{\text{initial position} > \lambda_v} \\
&= v\left(\frac{\lambda_v}{L} - \frac{v\tau}{L} \ln\left(1 + \frac{\lambda_v}{v\tau}\right)\right) + v\left(1 - \frac{\lambda_v}{L}\right) \\
&= v\left(1 - \frac{v\tau}{L} \ln\left(1 + \frac{\lambda_v}{v\tau}\right)\right).
\end{aligned}
\tag{16}
$$

As can be seen, in both cases the effective speed is lower than the walking speed $v$. Now, if $p(L)$ is the distribution of microtubule lengths, then the mean effective motor speed evaluated over the whole microtubule population becomes

$$
\langle v_{eff} \rangle = \int_0^\infty v_{eff}(L)\, p(L)\, dL.
\tag{17}
$$

We calculate this effective speed for each motor numerically using the measured distribution $p(L)$.

The end-residence time $\tau$ was measured for rat kinesin-1 motors to be $\approx 0.5\ s$ (*Belsham and Friel, 2019*). We take this estimate for our K401 motors (*D. melanogaster* kinesin-1) and since, to our knowledge, there is no available data on end-residence times for Ncd and Kif11 motors, we use the same estimate for them (we note that in simulation studies too the end-residence time is typically guessed *Surrey et al., 2001*).

Using this $\tau$ estimate, the measured distribution $p(L)$, and the motor speed ($v$) and processivity ($\lambda_v$) values from *Table 1*, we numerically evaluate the relative decrease in the effective speeds of the motors as

$$\text{K401:} \quad \langle v_{\text{eff}} \rangle / v \approx 0.71, \tag{18}$$

$$\text{Kif:} \quad \langle v_{\text{eff}} \rangle / v \approx 0.94, \tag{19}$$

$$\text{Ncd:} \quad \langle v_{\text{eff}} \rangle / v \approx 0.99 \, (0.71). \tag{20}$$

Here, we made two estimates for Ncd, first using its single-molecule processivity ($\approx 1$ step) reported in **Table 1**, and then the 100-fold increased processivity potentially reached due to collective effects mentioned in the main text. As a consequence of this effective speed reduction, we expect factors of $\approx 1.41$, $\approx 1.06$, and $\approx 1.01$ (1.41) increase in the inferred $\lambda_0$ values of K401, Kif11, and Ncd motors, respectively.

## Broader spread of the tubulin profile

In this section, we discuss the feature of a broader tubulin distribution in greater detail. To gain analytical insights, we first consider an idealized scenario where the motor profile can be represented as an exponential decay with a constant offset for the free motor population (**Appendix 1—figure 8a**). Such a scenario is approximately met for many of our measured motor profiles. Using our modeling framework, we find that the microtubule distribution corresponding to such a motor profile has the shape of a truncated sigmoid (**Appendix 1—figure 8b**, see the second part of this section for the derivation). Indeed, microtubule distributions resembling a sigmoidal shape are observed often in our asters (**Figure 2—figure supplement 1**), two examples of which are shown in **Appendix 1—figure 8c**.

One notable implication of this analytical connection between the two profiles is that microtubules necessarily have a broader distribution than motors, once the offset levels at the aster edge are subtracted off. To find whether this is a ubiquitous feature of our asters, we introduce radial distances $r_{1/2}^{(\text{m})}$ and $r_{1/2}^{(\text{t})}$ standing for the positions where the motor and tubulin distributions, respectively, are at their mid-concentrations (**Appendix 1—figure 8d**). The ratio $\left(r_{1/2}^{(\text{t})} - r_{\text{in}}\right)/\left(r_{1/2}^{(\text{m})} - r_{\text{in}}\right)$, if greater than 1, would then be an indicator of a wider tubulin profile. Calculating this ratio for all of our asters, we find that it is always greater than 1 for all motor types (**Appendix 1—figure 8e**), suggesting the generality of the feature.

### Idealized scenario with an exponentially decaying motor profile

Here, we first derive the analytical form for the tubulin distribution in the idealized scenario where the motor profile can be approximated as an exponential decay. We then demonstrate that, when normalized, this distribution is always broader than the motor distribution.

We start off by writing the motor distribution as

$$m_{\text{tot}}(r) \approx m_\infty + \Delta m \, e^{-(r - r_{\text{in}})/\lambda}, \tag{21}$$

where $\lambda$ is the decay length scale, $m_\infty$ is the background motor concentration corresponding to the free motor population, and $\Delta m$ is the amplitude of the exponential decay. Next, using **Equation 5** as well as the definition $m_{\text{tot}}(r) = m_{\text{b}}(r) + m_{\text{f}}(r)$, we set out to obtain the distributions of bound and free motor populations. From **Equation 5**, we have

$$\begin{aligned} m_{\text{b}}(r) &= -\lambda_0 \, m_{\text{f}}'(r) \\ &= -\lambda_0 \left( m_{\text{tot}}'(r) - m_{\text{b}}'(r) \right) \Rightarrow \end{aligned} \tag{22}$$

$$m_{\text{b}}'(r) - \frac{m_{\text{b}}(r)}{\lambda_0} = m_{\text{tot}}'(r) = -\frac{\Delta m}{\lambda} e^{-(r - r_{\text{in}})/\lambda}. \tag{23}$$

Solving for $m_{\text{b}}(r)$, we find

$$m_{\text{b}}(r) = C_{\text{b}} \, e^{(r - r_{\text{in}})/\lambda_0} + \Delta m \, \frac{\lambda_0}{\lambda + \lambda_0} e^{-(r - r_{\text{in}})/\lambda}, \tag{24}$$

where $C_{\text{b}}$ is an integration constant. Because the approximation **Equation 21** applies to a finite radial interval $r \in [r_{\text{in}}, r_{\text{out}}]$, the constant $C_{\text{b}}$ is generally nonzero. It specifies the relative contributions of free and bound motor populations to the total motor distribution $m_{\text{tot}}(r)$.

The free motor population is found by simply subtracting **Equation 24** from **Equation 21**, that is,

$$m_{\text{f}}(r) = m_{\text{tot}}(r) - m_{\text{b}}(r)$$
$$= m_\infty + \Delta m \frac{\lambda}{\lambda + \lambda_0} e^{-(r - r_{\text{in}})/\lambda} - C_{\text{b}}\, e^{(r - r_{\text{in}})/\lambda_0}. \tag{25}$$

Having obtained expressions for the two motor populations (bound and free), we now recall *Equation 6* that relates these two populations through the local tubulin density. Using *Equation 6*, we find the tubulin density as

$$
\begin{aligned}
\rho_{\text{MT}}(r) &= K_{\text{d}} \frac{m_{\text{b}}(r)}{m_{\text{f}}(r)} \\
&= K_{\text{d}} \frac{\Delta m \frac{\lambda_0}{\lambda + \lambda_0} e^{-(r - r_{\text{in}})/\lambda}}{m_\infty + \Delta m \frac{\lambda}{\lambda + \lambda_0} e^{-(r - r_{\text{in}})/\lambda}} \\
&= K_{\text{d}} \frac{\lambda_0}{\lambda} \frac{e^{-(r - r_{\text{in}})/\lambda}}{\frac{m_\infty}{\Delta m}\left(1 + \lambda_0/\lambda\right) + e^{-(r - r_{\text{in}})/\lambda}} \\
&= K_{\text{d}} \frac{\lambda_0}{\lambda} \frac{e^{-(r - r_{\text{in}})/\lambda}}{\gamma + e^{-(r - r_{\text{in}})/\lambda}},
\end{aligned} \tag{26}
$$

where we introduced the effective parameter $\gamma \equiv (m_\infty/\Delta m)(1 + \lambda_0/\lambda)$. *Equation 26* represents a partial sigmoid, the precise shape of which in the $r > r_{\text{in}}$ region is defined through the parameter $\gamma$ (*Appendix 1—figure 8b*).

To formally demonstrate that the tubulin profiles predicted in *Equation 26* are necessarily broader than the motor profile, we first normalized them after subtracting off the concentration values at the outer boundary, namely,

$$
\begin{aligned}
\hat{m}_{\text{tot}}(r) &= \frac{m_{\text{tot}}(r) - m_{\text{tot}}(r_{\text{out}})}{m_{\text{tot}}(r_{\text{in}}) - m_{\text{tot}}(r_{\text{out}})} \\
&= \frac{\Delta m\, e^{-(r - r_{\text{in}})/\lambda} - \Delta m\, e^{-(r_{\text{out}} - r_{\text{min}})/\lambda}}{\Delta m - \Delta m\, e^{-(r_{\text{out}} - r_{\text{in}})/\lambda}} \\
&= \frac{e^{-(r - r_{\text{in}})/\lambda} - e^{-(r_{\text{out}} - r_{\text{in}})/\lambda}}{1 - e^{-(r_{\text{out}} - r_{\text{in}})/\lambda}},
\end{aligned} \tag{27}
$$

$$
\begin{aligned}
\hat{\rho}_{\text{MT}}(r) &= \frac{\rho_{\text{MT}}(r) - \rho_{\text{MT}}(r_{\text{out}})}{\rho_{\text{MT}}(r_{\text{in}}) - \rho_{\text{MT}}(r_{\text{out}})} \\
&= \frac{\frac{e^{-(r - r_{\text{in}})/\lambda}}{\gamma + e^{-(r - r_{\text{in}})/\lambda}} - \frac{e^{-(r_{\text{out}} - r_{\text{in}})/\lambda}}{\gamma + e^{-(r_{\text{out}} - r_{\text{in}})/\lambda}}}{\frac{1}{\gamma + 1} - \frac{e^{-(r_{\text{out}} - r_{\text{in}})/\lambda}}{\gamma + e^{-(r_{\text{out}} - r_{\text{in}})/\lambda}}} \\
&= \frac{(\gamma + 1)(\gamma + e^{-(r_{\text{out}} - r_{\text{in}})/\lambda})}{(\gamma + e^{-(r - r_{\text{in}})/\lambda})(\gamma + e^{-(r_{\text{out}} - r_{\text{in}})/\lambda})} \times \frac{\gamma e^{-(r - r_{\text{in}})/\lambda} - \gamma e^{-(r_{\text{out}} - r_{\text{in}})/\lambda}}{\gamma - \gamma e^{-(r_{\text{out}} - r_{\text{in}})/\lambda}} \\
&= \frac{\gamma + 1}{\gamma + e^{-(r - r_{\text{in}})/\lambda}} \times \underbrace{\frac{e^{-(r - r_{\text{in}})/\lambda} - e^{-(r_{\text{out}} - r_{\text{in}})/\lambda}}{1 - e^{-(r_{\text{out}} - r_{\text{in}})/\lambda}}}_{\hat{m}_{\text{tot}}(r)} \\
&= \frac{\gamma + 1}{\gamma + e^{-(r - r_{\text{in}})/\lambda}} \times \hat{m}_{\text{tot}}(r).
\end{aligned} \tag{28}
$$

The local ratio of normalized tubulin and motor densities then becomes

$$\frac{\hat{\rho}_{\text{MT}}(r)}{\hat{m}_{\text{tot}}(r)} = \frac{\gamma + 1}{\gamma + e^{-(r - r_{\text{in}})/\lambda}} > 1, \tag{29}$$

which is always greater than 1 in the $r > r_{\text{in}}$ region. This is indicative of the shoulder that the normalized tubulin profile often forms over normalized motor profile and demonstrates the broader spread of the tubulin distribution in this idealized setting.

## Relative widths of the two distributions

We can see from *Appendix 1—figure 8e* that the relative widths of the motor and microtubule distributions differ most for Ncd motors (median ratio $\approx 1.55$), while for K401 and Kif, the widths are more comparable (median ratio $\approx 1.35$ for both motors). Here, we offer an explanation for this difference between the motors using the analytical insights developed earlier in the section.

Specifically, *Equation 29* and *Appendix 1—figure 8b* suggest that lower values of $\gamma$ correspond to broader microtubule distributions. Substituting $r = r_{\text{in}}$ in *Equation 26*, we can write $\gamma$ as

$$\gamma = \frac{1}{\tilde{\rho}_0} \frac{\lambda_0}{\lambda} - 1, \tag{30}$$

where $\tilde{\rho}_0 = \rho_{MT}(r_{\text{in}})/K_{\text{d}}$ is the microtubule concentration near the core in units of $K_{\text{d}}$.

From the concentration profiles in *Figure 2—figure supplement 1*, we can estimate the motor decay length scale to be $\lambda \approx 20, 15, 10$ μm for K401, Kif11, and Ncd, respectively. Then, from our model fitting procedure, we inferred $\lambda_0 \approx 40, 15, 20$ μm, resulting in length scale ratios $\lambda_0/\lambda \approx 2, 1, 2$ for the three motors. Lastly, again inspecting the profiles in *Figure 2—figure supplement 1*, we find the microtubule concentrations near the core in $K_{\text{d}}$ units to be $\tilde{\rho}_0 \approx 0.5, 0.2, 1.3$. Note that the microtubule concentration near the core (in $K_{\text{d}}$ units) is the highest for Ncd. In the final step, we substitute these estimated values for $\lambda_0/\lambda$ and $\tilde{\rho}_0$ into *Equation 30*, and evaluate $\gamma$ for the K401, Kif11, and Ncd, respectively, to be $\gamma \approx \{3, 4, 0.3\}$.

This matches well with the intuition of our simple analytical study (*Appendix 1—figure 8b*) and the observed ratios of distribution widths (*Appendix 1—figure 8e*). Namely, large values of $\gamma$ for K401 and Kif11 ($\approx 3$ and 4, respectively) suggest a closer correspondence between the normalized motor and microtubule profiles, while the lower values of $\gamma$ for Ncd ($\approx 0.3$) suggests a wider microtubule distribution, as was observed in our asters.

## Merger analysis

To measure contractile speeds in a pseudo one-dimensional network, we performed aster merger experiments. In these experiments, two asters were formed with 50 μm diameter excitation disks, at varying initial separations (either 200, 400, 600, 800, or 1000 μm). The two disks were illuminated for 30 or 60 frames for experiments with Kif11 (15 s between frames). After 30 or 60 frames, a thin rectangle was illuminated between the two asters. This caused the formation of a network which contracted, pulling the two asters together. An image of two asters connected in this way is shown in *Figure 3*.

We used two methods to measure contractile rates. One was using optical flow to measure speeds throughout the network, as shown in *Figure 3b*. The open source package, Open CV, was used for the optical flow measurements. We used a dense optical flow measurement using cv.CalcOpticalFlowFarneback, which calculates optical flow speeds using the Gunnar Farneback algorithm.

To calculate the maximum contractile rate, we tracked the positions of the asters over time. The microtubule fluorescence in the region containing the asters and network (pixels 450–650 in the y-direction) was summed across the y-dimension to make a one-dimensional line trace of fluorescence (*Appendix 1—figure 10a*). This trace was smoothed using a butter filter to reduce the noise in the fluorescence intensity. An example smoothed line trace is shown in *Appendix 1—figure 10a*. The asters were then identified as the large peaks in fluorescence intensity using scipy.signal.find_peaks. The identified peaks are shown as red dots in *Appendix 1—figure 10*. The corresponding image that the line trace is from is shown in *Appendix 1—figure 10b* with a 50×50 pixel box around the identified aster colored in black. From the identified aster coordinates, the distance between the asters was calculated and the difference between distances in successive frames was used to calculate the merger speed. The reported speed is the speed of a single aster. Thus, the speed calculated from the change in distance between asters is divided by two, assuming the asters are moving at equal speed.

Example traces of calculated aster speeds over time after the beginning of illuminating the bar region is shown in *Appendix 1—figure 10c*. The speed of the asters increases while the network connecting them forms, peaks, and then decreases as the asters near each other. The peak speed is reported as the maximum aster speed in the main text. These maximum aster speeds as a function of initial separation were fit to a line using scipy.optimize.curve_fit to determine the slope of the increase in speed with separation. The best fit lines determined with this method are plotted along with the data for Ncd and Kif11 in *Figure 3c*. The fit slope for Ncd was 0.0024 and 0.0013 s$^{-1}$ for Kif11. The ratio (Ncd/Kif11) of these slops is $\approx 1.8$. In comparison, the ratio between measured motor speeds (115 nm/s for Ncd and 70 nm/s for Kif11) is $\approx 1.6$. The slope calculated for K401 was $\approx 0.0043 \, \text{s}^{-1}$. Thus, the ratio of slopes of aster speeds versus separation is in good agreement with the ratio of motor speeds, suggesting that the motor speed sets this slope. To illustrate this point, the best fit lines shown in *Figure 3c* are plotted again, with the

slopes divided by the single-motor speed. In this way, the three lines now overlap, as shown in *Appendix 1—figure 9*.

## Model of network contraction

In our aster merger experiments, we observed a linear relationship between distance from the center of the network and aster speed. This is seen in the plot of x-velocity versus position in *Figure 3b* and also in the plot of maximum aster speed versus initial separation in *Figure 3c*. Juniper et al. observed a similar relationship between position from the center of the network and contraction speed during aster formation in *Juniper et al., 2018*. The speed measured at the ends of the network is a simple summation of the velocities of walking motors, the same way that a person appears to walk faster on a moving walkway than on a stationary walkway. In an ideal case, each pair of microtubules would contract relative to each other, meaning that the speed measured at the end of the network would be the number of microtubules multiplied by the walking speed of each motor. There are a number of factors that make our experiments different from this ideal case, for example, the microtubules are not perfectly aligned and there are some dead motors in the system. Thus, we consider the contractile region, of length $L$, to be composed of $N$ contractile units, which are bundles of microtubules with characteristic length $\ell$, that contract relative to each other. The velocity measured at the ends of the network, then is $\nu$ times the number of contractile units ($L/\ell$). We summarize this in *Figure 3—figure supplement 1*.

$$V \sim N\nu_m \qquad V \sim (L/\ell)\nu_m \qquad (31)$$

where $V$ is the network contractile rate and $\nu_m$ is the motor speed. Our observed relationship that the ratio of the slope of speed increase matched the ratio of single-motor speeds (as discussed in the main text) supports this hypothesis. Further, we can estimate that $L \approx 50$–$100$ µm, indicating that the characteristic length scale is larger than a single microtubule.

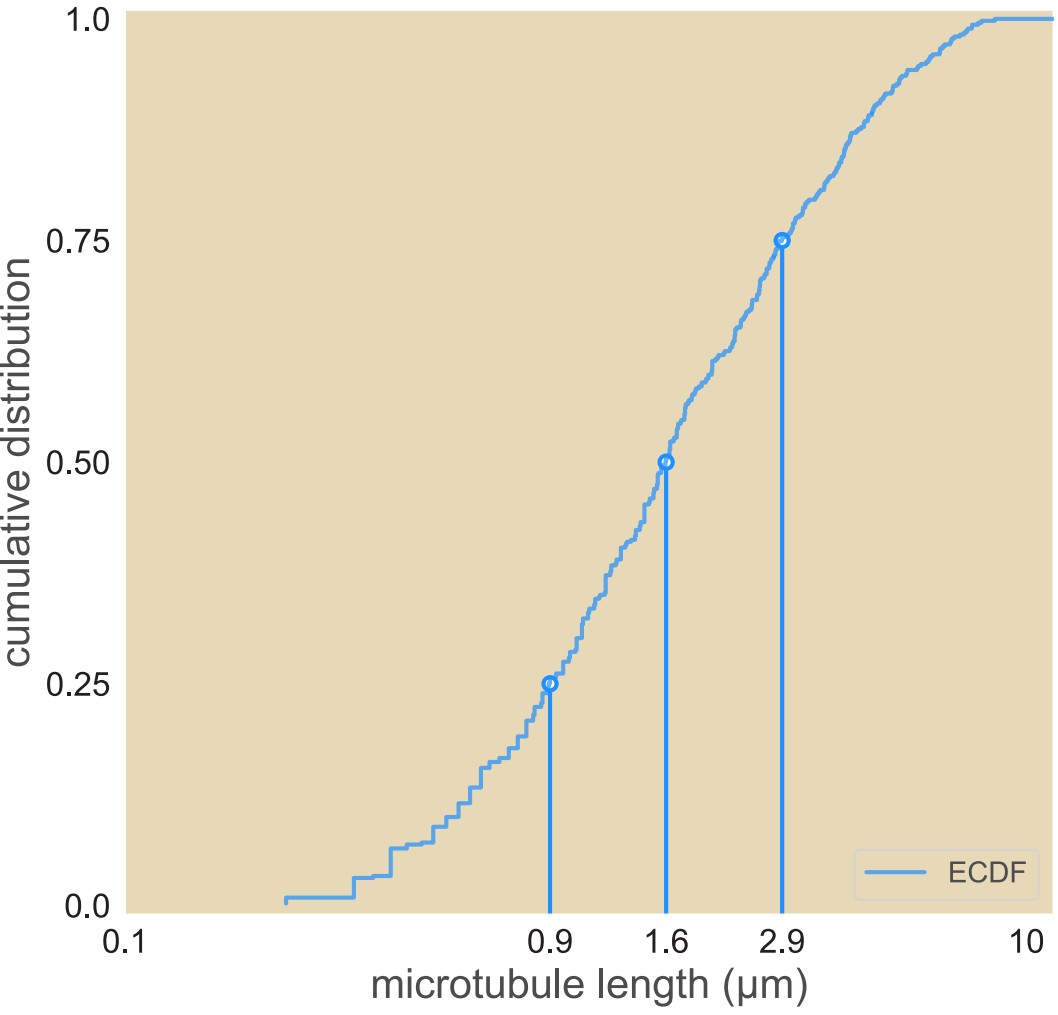

**Appendix 1—figure 1.** Cumulative distribution of microtubule lengths. The 25%, 50%, and 75% length are marked.

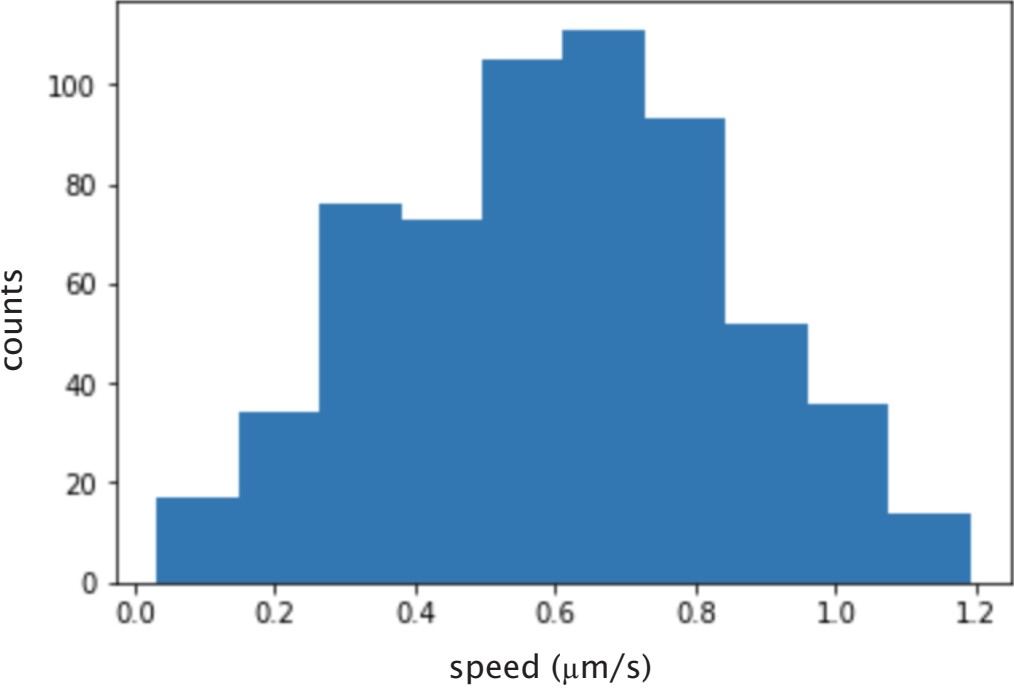

**Appendix 1—figure 2.** Histogram of calculated instantaneous speed of microtubules glided by K401 motors. The mean speed is $\approx 600$ nm/s.

a)                                                        b)

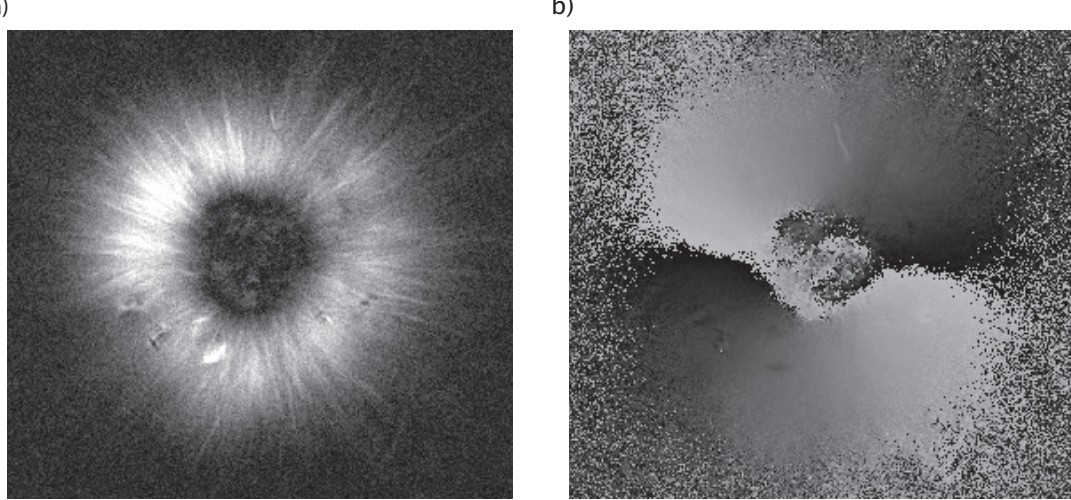

**Appendix 1—figure 3.** Aster centers are disordered and the arms are aligned radially. (**a**) Retardance image of an aster taken with a Pol-Scope. The arms of the aster result from their alignment and the magnitude of retardance is proportional to the number of microtubules in a bundle. The dark center indicates no alignment of microtubules in that region. (**b**) Azimuthal angle of microtubule alignment in an aster. Black is 0 and white is 180.

**FLUORESCENCE RECOVERY IN A STEADY–STATE ASTER**

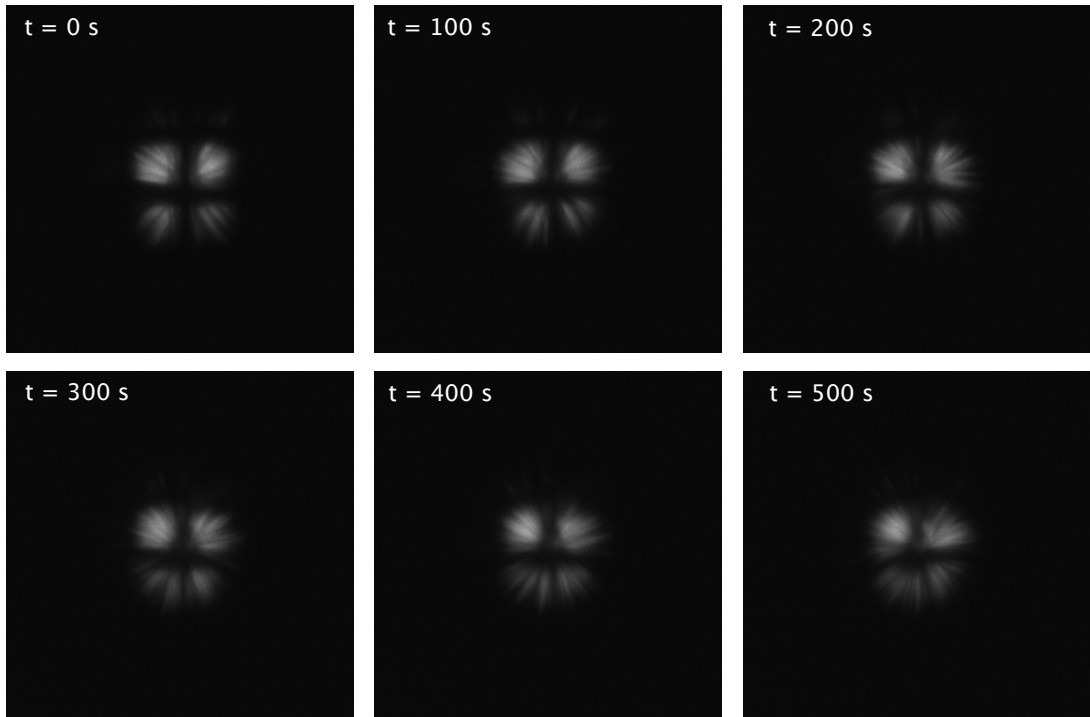

**Appendix 1—figure 4.** Fluorescence recovery of the microtubules in a steady-state aster. Images of the microtubule fluorescence after photobleaching in a grid pattern of an aster formed with Ncd236 motors.

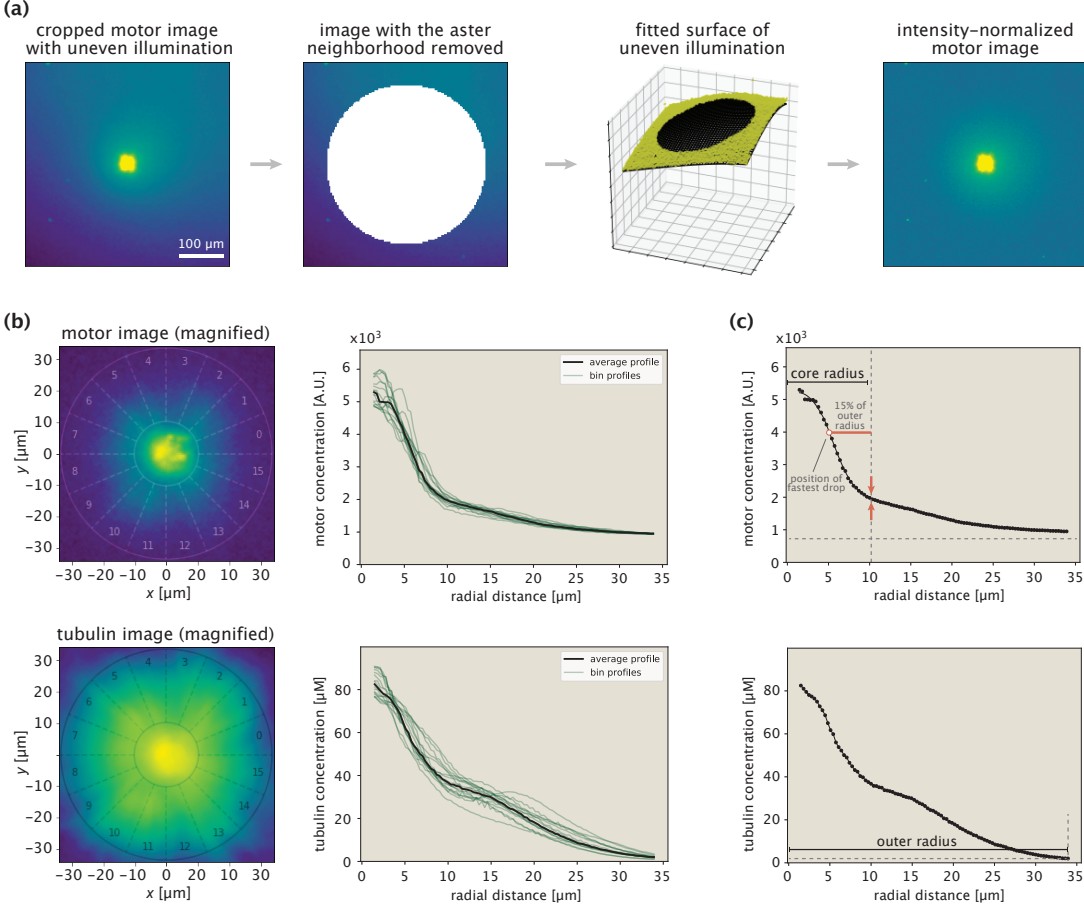

**Appendix 1—figure 5.** Procedure for extracting protein concentration profiles demonstrated on an example aster. (**a**) Steps taken in normalizing the fluorescence of motor images. The immediate aster region is shown with a saturated color to make it possible to see the nonuniform background fluorescence. (**b**) Aster center identification and extraction of radial concentration profiles. The numbers indicate the wedges at different angular positions. The two circles in the images indicate the inner and outer bounds. (**c**) Determination of inner and outer bounds based on the motor and tubulin profiles, respectively.

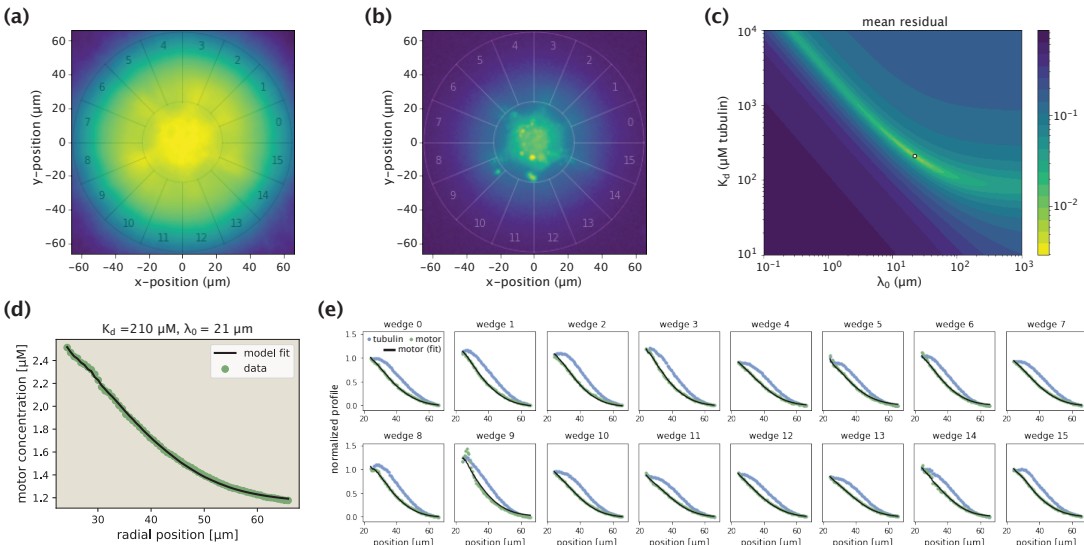

**Appendix 1—figure 6.** Demonstration of the model fitting procedure for average as well as separate wedge profiles. (**a,b**) Fluorescence images of an example Kif11 aster in tubulin (**a**) and motor (**b**) channels. Sixteen different

*Appendix 1—figure 6 continued on next page*

*Appendix 1—figure 6 continued*

wedges are separated and numbered. (**c**) Landscape of fit residuals when varying the two effective parameters $K_d$ and $\lambda_0$. For each pair, an optimal scaling coefficient $C$ is inferred before calculating the residual. The dot at the brightest spot stands for the optimal pair (or, the arrow indicates the location of the optimal pair in the landscape). (**d**) Average motor profile and the model fit, along with the inferred parameters. (**e**) Collection of fits to separate wedge profiles using the optimal $(K_d, \lambda_0)$ pair inferred from the average profile.

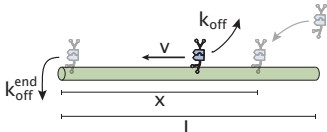

**Appendix 1—figure 7.** Schematic representation of initial motor binding, advection, and stalling at the microtubule end.

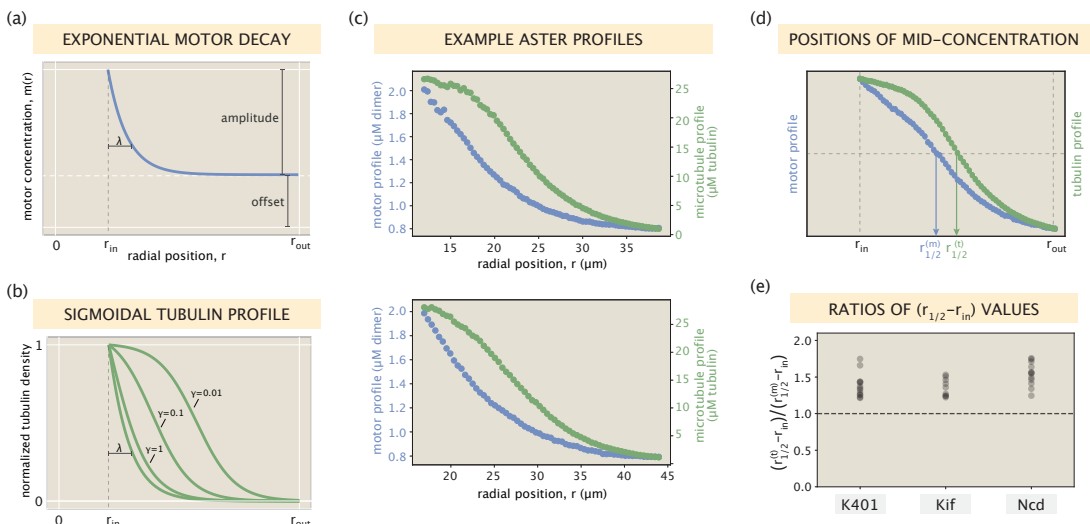

**Appendix 1—figure 8.** Relationship between motor and microtubule distributions. (**a**) An idealized exponentially decaying motor profile with a constant offset. (**b**) Set of sigmoidal tubulin profiles corresponding to the exponentially decaying motor profile. The precise curve depends on the shape parameters of the motor profile and the motor type via an effective constant $\gamma$ (see SI section 'Broader spread of the tubulin profile for details). (**c**) Two example profiles from Ncd asters that resemble the setting in panels (**a**) and (**b**). Blue and green dots represent measured concentrations of motors and microtubules, respectively. (**d**) Radial positions in the $[r_{in}, r_{out}]$ interval where the motor and tubulin concentrations take their middle values. (**e**) The ratio $\left(r_{1/2}^{(t)} - r_{in}\right)/\left(r_{1/2}^{(m)} - r_{in}\right)$ calculated for all of the aster profiles.

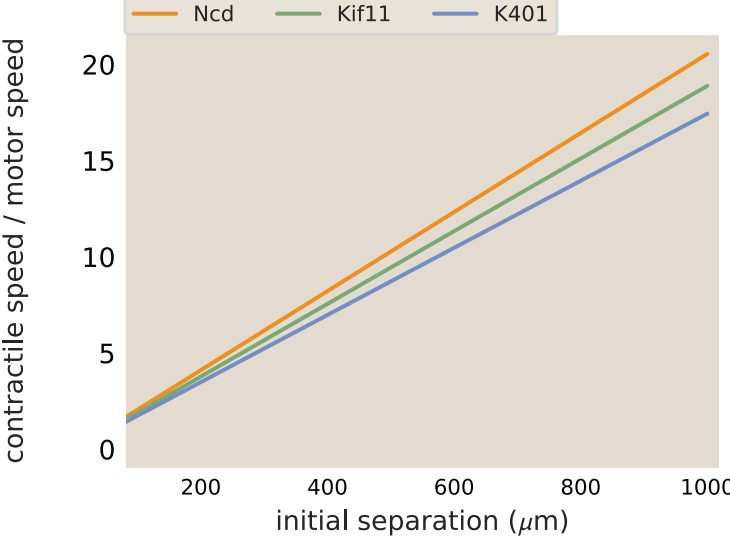

**Appendix 1—figure 9.** Merger speeds depend on motor stepping speed. The best fit lines from the contraction rate measured in aster mergers are plotted, normalized by the measured motor speed.

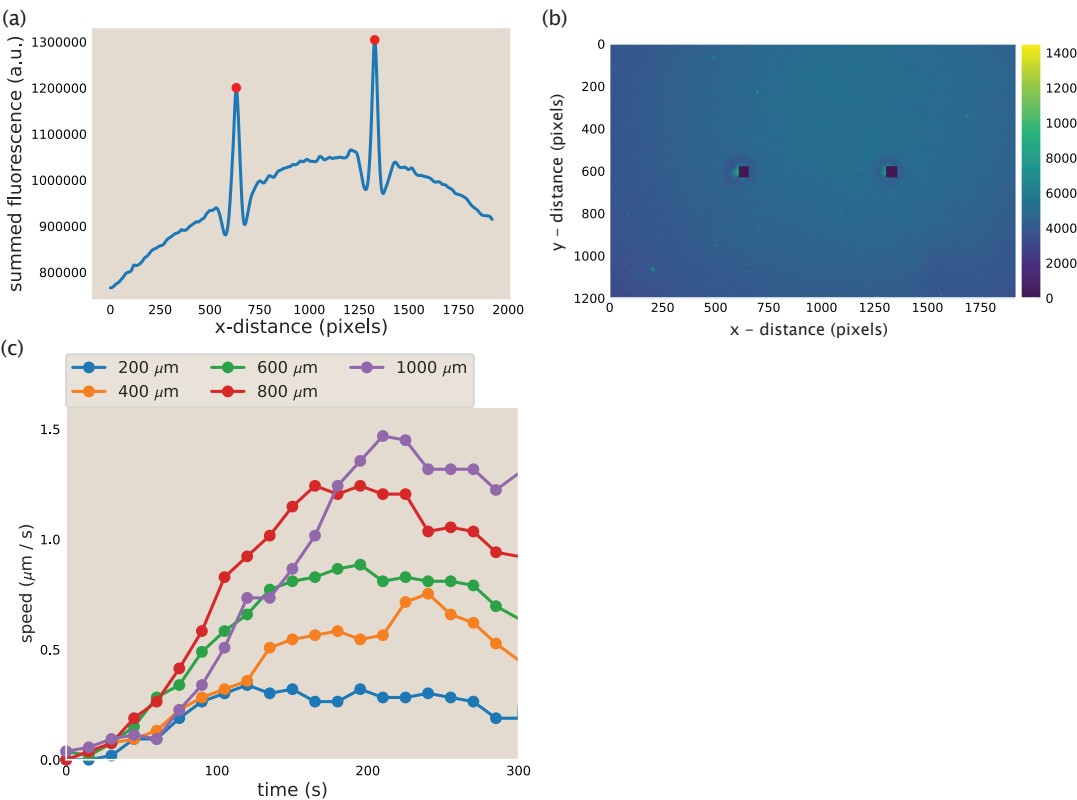

**Appendix 1—figure 10.** Aster identification and speed calculation in aster mergers. (**a**) Example y-summed microtubule fluorescence from an image during aster merger. The peaks represent the asters and the red dots are the position of the asters as identified by the code. (**b**) Image corresponding to the fluorescence plotted in (**a**). The blacked out squares are the location of the asters identified from the peaks in (**a**). (**c**) Measured aster speed versus time during aster merger. Each dot is a single measured speed and the colors represent the initial separation of the asters.

