## [Editor Report]

Banks et al. demonstrate that the organization and dynamics of microtubule/kinesin asters depend upon the speed and processivity of motors. By combining in vitro reconstitutions with theory, they are able to extract parameters that relate to the dynamics of the motors. This study is of interest to readers working on microtubules, motors, and in the active matter physics field.

---

## [Decision Letter]

**Decision letter after peer review:**

Thank you for submitting your article "Motor processivity and speed determine structure and dynamics of motor-microtubule assemblies" for consideration by *eLife*. Your article has been reviewed by 3 peer reviewers, one of whom is a member of our Board of Reviewing Editors, and the evaluation has been overseen by Anna Akhmanova as the Senior Editor. The reviewers have opted to remain anonymous.

Essential revisions:

1) Several of your observations seem to be at odds with previous results. Specifically, it has been shown that cooperative effects between kinesin-1 do not increase their walking speed. You find that the cooperativity of the motors (including kinesin-1) leads to increased speed of two merging asters. It is not clear from the experimental or theoretical data how the contractile speeds of two merging asters can move up to 4-times faster than an individual motor protein. In the text above you mentioned that processivity and speed might be hindered by obstacles and crowding of motors, why is then the speed so high? Previous studies have shown that increasing the number of kinesins on a bead does not increase the speed of the bead. How does this result fit to your observation? Please, comment on these points in your revised manuscript.

2) Some questions came up about the general behavior of the aster organization. Why does only one and not multiple asters form if the region of excitation is over 10x the size? Furthermore, a much larger excitation diameter than the size of the resultant structure suggests the amount of dimeric motor is not limiting. Why then does the size of the aster increase with excitation diameter? Please, comment on these questions in your revised manuscript.

3) Asters sizes that vary with speed/processivity likely vary in their connectivity and thereby viscoelasticity. You draw analogies to contractile units, presumably in analogy to muscle. Are viscoelastic materials with variable processivity-dependent connectivity consistent with the picture of elastic, well-connected contractile units? Comment on the geometric/mechanical pre-requisites for this comparison.

*Reviewer #1 (Recommendations for the authors):*

The dynamics of contractile microtubule-kinesin and actomyosin networks has been studied previously and yielded the conclusion that the contraction speed increased towards the boundaries which correspond to the aster cores in the present work. The authors might consider mentioning this in their manuscript.

*Reviewer #2 (Recommendations for the authors):*

– "Thus, we conclude that the rate of contraction in the network is set by the motor speed and the increase of network speed with distance is due to adding more connected contractile units." Do you have an experimental or theoretical approach to test you hypothesis? If contraction is set by the motor speed should the maximum contraction speed not equal the speed of a single motor? In the text above you mentioned that processivity and speed might be hindered by obstacles and crowding of motors, why is than the speed so high? Previous studies have shown that increasing the number of kinesins on a bead does not increase the speed of the bead. How does this result fit to your observation?

– At steady-state of the system the microtubules have on average no radial movement. Did you ever perform a FRAP of the aster? I would like to know if the microtubules in the aster at steady state are still dynamic, and if the dynamic properties change before and at steady-state. Same for the motors.

– Between the three different motor-induced asters is there a different behavior of the aster at steady-state and before? What is the timing to reach steady-state for the asters with the different motors? Are the dynamic properties of the asters different? Is the nucleation of the aster different?

– "While the inferred values for the two slower motors are well within the order-of-magnitude of our guess, the inferred λ0 for the faster K401 motor is much higher than what we anticipated. " You explain this by Kinesin-1 stalling at the microtubule end, however Kinesin-1 is not known to stall at the microtubule end, but just falls of without stopping. Further you consider that the median MT length is 1.6µm, but your mean length is around 7 µm. Do you have any alternative explanation why K401 inferred λ0 is so much higher? Please, could also add a size distribution blot of the stabilized microtubules used for the experiments?

– Figure 1C: Multiple peaks of FI are visible. Are there periodic rings of high tubulin intensity forming? Do they cluster with the motor concentration? Is the number of peaks dependent on the aster size, or the motor?

*Reviewer #3 (Recommendations for the authors):*

The authors succeed in demonstrating how the microscopic properties of motors can determine large-scale material structure. This is the putative principal impact of the work. However, some of these results have been seen in other motor/filament systems, such as actomyosin. An example is the relationship between contractile velocity and the dimension of the assembly (e.g. Thoresen, BJ, 2011, Linsmier et al., Nat Comm, 2016, Schuppler, Nat Comm, 2016). While similar relationships observed in microtubule /kinesin systems are notable, due to differences in motor proteins parameters, filament mechanics, etc, these prior works should be mentioned in light of the contributions of the present work.

---

## [Author Response]

Essential revisions:1) Several of your observations seem to be at odds with previous results. Specifically, it has been shown that cooperative effects between kinesin-1 do not increase their walking speed. You find that the cooperativity of the motors (including kinesin-1) leads to increased speed of two merging asters. It is not clear from the experimental or theoretical data how the contractile speeds of two merging asters can move up to 4-times faster than an individual motor protein. In the text above you mentioned that processivity and speed might be hindered by obstacles and crowding of motors, why is then the speed so high? Previous studies have shown that increasing the number of kinesins on a bead does not increase the speed of the bead. How does this result fit to your observation? Please, comment on these points in your revised manuscript.

We thank the editor for these comments. We believe that the increase in speed we observe in aster merger experiments is not due to cooperativity between motors but rather is a geometric effect. Contraction between two asters is driven by a series of microtubules connected by kinesin motors, that each walk at a characteristic speed, *ν*. The speed measured at the ends of the network is a simple sum of the velocities of walking motors, the same way that a person appears to walk faster on a moving walkway than on a stationary walkway. In an ideal case, each pair of microtubules would contract relative to each other, meaning that the speed measured at the end of the network would be the number of microtubules multiplied by the walking speed of each motor. There are a number of factors that make our experiments different from this ideal case, for example the microtubules are not perfectly aligned and there are some dead motors in the system. Thus, we consider the contractile region, of length L, to be composed of N contractile units, which are bundles of microtubules with characteristic length ℓ, that contract relative to each other. The velocity measured at the ends of the network, then is *ν* times the number of contractile units (L/ℓ).

We added the above explanation (lines 722-732) as well as as Figure 3—figure supplement 3 to the supplemental information of the manuscript.

As reviewer #3 pointed out, this finding is in agreement with previous results on actomyosin networks, which found that contractile rates increase linearly towards the ends of the network (Thoresen, et al. 2022; Linsmeier, et al. 2016; Schuppler, at al. 2016). These works proposed a ‘telescoping’ model of contraction, which is in agreement with our findings for kinesinmicrotubule contractile networks. We added reference to these previous findings in lines 249252 as follows:

“These findings are in agreement with results from several studies of contractile rates in actomyosin networks, that suggested 'telescoping' models of contraction, and suggest that this may be a common mechanism across cytosketetal networks [32-34].”

2) Some questions came up about the general behavior of the aster organization. Why does only one and not multiple asters form if the region of excitation is over 10x the size? Furthermore, a much larger excitation diameter than the size of the resultant structure suggests the amount of dimeric motor is not limiting. Why then does the size of the aster increase with excitation diameter? Please, comment on these questions in your revised manuscript.

We believe that one aster is formed even with a large excitation region due to high connectivity between the microtubules in our system. We are careful in the experiments in this manuscript to remain in a regime of microtubule and motor concentrations that results in a single aster. However, by altering the system, with different concentrations of motors, microtubules, and crowding agent (glycerol, in our case), different results can be achieved. For example, in some cases, we can see multiple asters after illumination of a region, and in others, the contractile network does not fully separate from the background, which can lead to inward flows of microtubules into the contractile region. This warrants further investigation in the future. We added SI section 8.10 (lines 418-425) and Figure 1—figure supplement 3 to clarify this and reference to them in the main text, lines 78-80, as follows:

“For the purposes of this study, we were careful to remain in a regime of motor and microtubule concentrations that produced a single aster upon illumination. However, by varying concentrations of the motors and microtubules, it is possible to form multiple smaller asters within the region, a few examples of which are shown in Figure 1—figure supplement 3. How varying the composition of the reaction mixture impacts the resulting structures warrants further investigation.”

The increase in aster size with excitation diameter was also reported in our prior work (Ross, et al. 2019), in which we only used K401 motors. In this work, we reported that the aster size appears to scale with the volume of the excitation region. Thus, it seems that there is a density limit of the microtubules that leads to the increase in aster size with excitation diameter. This density limit could be due to steric interactions between the microtubules

3) Asters sizes that vary with speed/processivity likely vary in their connectivity and thereby viscoelasticity. You draw analogies to contractile units, presumably in analogy to muscle. Are viscoelastic materials with variable processivity-dependent connectivity consistent with the picture of elastic, well-connected contractile units? Comment on the geometric/mechanical pre-requisites for this comparison.

The editor is correct that the system is viscoelastic; as is characteristic of viscoelastic materials, the properties depend on the timescale looked at. In this manuscript, the contraction rates that we report are from the initial contraction of the network, the first 100-200 seconds. We believe that during this timeframe, the network can be thought of as elastic, since the relaxation timescale should be much longer. While the optogenetic proteins unbind on the order of 20 seconds, we believe there are multiple motors crosslinking any two microtubules together. Thus, the relaxation timescale would be the time for most, if not all of these motors to unbind. On longer time scales, the network properties differ from an elastic network due to the binding/unbinding of motors and exchange of bonds between the optogenetic proteins.

To address this point, we added sentences in lines 266-274, as follows:

“It is important to note that we only measure the initial contraction rate in these experiments, over the first ≈ 100 seconds. On this time scale, we think of the network as an elastic material, since the time for motor unbinding and rearrangement should be longer than this. The time for a single dimer to unbind is about 20 seconds, and for multiple dimers to unbind will be on average much longer. Thus, on the ≈ 100 seconds timescale we measure the speed over, the crosslinks can be considered roughly constant and we only observe the elastic properties of the system. The viscous properties will be observed on longer timescales, for the motors to unbind or optogenetic links to rearrange. Therefore, to account for the change in contraction rate throughout the process, rather than just the initial contraction rate, one would likely need to account for these viscous effects.”

Reviewer #1 (Recommendations for the authors):The dynamics of contractile microtubule-kinesin and actomyosin networks has been studied previously and yielded the conclusion that the contraction speed increased towards the boundaries which correspond to the aster cores in the present work. The authors might consider mentioning this in their manuscript.

We thank the reviewer for pointing out these previous works. We added reference to a few works that studied contraction of actomyosin networks and found that the dynamics are consistent with a ‘telescoping’ model, similar to our findings in lines 249-252:

“These findings are in agreement with results from several studies of contractile rates in actomyosin networks, that suggested 'telescoping' models of contraction, and suggest that this may be a common mechanism across cytosketetal networks [32-34].”

Reviewer #2 (Recommendations for the authors):– "Thus, we conclude that the rate of contraction in the network is set by the motor speed and the increase of network speed with distance is due to adding more connected contractile units." Do you have an experimental or theoretical approach to test you hypothesis? If contraction is set by the motor speed should the maximum contraction speed not equal the speed of a single motor? In the text above you mentioned that processivity and speed might be hindered by obstacles and crowding of motors, why is than the speed so high? Previous studies have shown that increasing the number of kinesins on a bead does not increase the speed of the bead. How does this result fit to your observation?

The reviewer is correct that single-molecule studies have demonstrated that multiple kinesins on a bead do not increase the speed of the bead for some kinesins (including K401). Single molecule studies have shown that for other motors, including Ncd, the speed of the bead can increase with the number of motors, up to a limit (Furuta, et al. 2013). Nonetheless, we attribute the increase in contractile speed with network size is not due to this effect, but rather due to ‘telescoping’ by adding contractile units to the network, as has been observed previously for actomyosin networks (Thoresen, et al. 2022; Linsmeier, et al. 2016; Schuppler, at al. 2016) and in this system, reported in Ross, et al. 2019 (see new figure 1—figure supplement 3).

In this work, we were interested in what determines the scale of the contractile speeds, and hypothesized that it depends on the single motor speed. The data reported here support this hypothesis, as we demonstrated that the slope of the increase in contraction speed with network size is proportional to the single motor speeds we measured through gliding assays. We made adjustments to the text to clarify this point, as summarized by the response to the first editor question.

– At steady-state of the system the microtubules have on average no radial movement. Did you ever perform a FRAP of the aster? I would like to know if the microtubules in the aster at steady state are still dynamic, and if the dynamic properties change before and at steady-state. Same for the motors.

We thank the reviewer for asking about the microtubule and motor dynamics. We have performed FRAP experiments on a steady state aster to investigate this. We photobleached the microtubules in a grid pattern in order to assess how they deform and rearrange over time, which is the subject of a future work. From these experiments, we observe that there is indeed no radial flux of microtubules once the aster is no longer changing, but they are still dynamic, as some angular motion can be seen. Before steady state, there is both advection and diffusion of microtubules, which we are looking further into in another study. We have also attempted FRAP experiments with the motors, but they recovered too quickly for us to make measurements with our current microscope setup.

We added a supplementary section and figure including images from a FRAP experiment of a steady state aster, section 8.12 (lines 443-449) and Figure 1—figure supplement 5. We added reference to this figure in lines 154-157, as follows:

“To assess the validity of this assumption, we performed FRAP experiments of steady-state asters, and observe little radial flux of the microtubules, an example is shown in Figure S1—figure supplement 5. They are still dynamic, as can be seen by the angular motion that leads to the recovery of fluorescence in the photobleached areas.”

– Between the three different motor-induced asters is there a different behavior of the aster at steady-state and before? What is the timing to reach steady-state for the asters with the different motors? Are the dynamic properties of the asters different? Is the nucleation of the aster different?

We thank the reviewer for their consideration, there are indeed some differences between the motors in how they reach the steady state aster. First, the dynamics are different, similar to the aster merger experiments. Kinesin-1 takes the least amount of time to form an aster, followed by Ncd and Kif11 is the slowest; this is in line with the single motor speeds.

We added explanation of this in the main text, lines 104-106, as follows:

“Interestingly, the dynamics of aster formation by these motors seemed to roughly scale with the motor speeds – K401 formed asters the quickest, followed by Ncd236, and Kif11 took the most time to form an aster.”

– "While the inferred values for the two slower motors are well within the order-of-magnitude of our guess, the inferred λ0 for the faster K401 motor is much higher than what we anticipated. " You explain this by Kinesin-1 stalling at the microtubule end, however Kinesin-1 is not known to stall at the microtubule end, but just falls of without stopping. Further you consider that the median MT length is 1.6µm, but your mean length is around 7 µm. Do you have any alternative explanation why K401 inferred λ0 is so much higher? Please, could also add a size distribution blot of the stabilized microtubules used for the experiments?

We believe that the reviewer is referring to the microtubule length reported in our earlier work (Ross, et al. 2019). The microtubules used in the current work have an average length of 1.6 µm. There is a plot of the size distribution of the microtubules used in this work in SI section 8.4, Figure 1—figure supplement 1.

We are not aware of studies reporting that Kinesin-1 just falls off at the end. As far as we are aware, from simulation results, it is generally required for Kinesin-1 and other motors to stall at microtubule ends in order to form an aster, such as in Surrey, et al., 2001.

– Figure 1C: Multiple peaks of FI are visible. Are there periodic rings of high tubulin intensity forming? Do they cluster with the motor concentration? Is the number of peaks dependent on the aster size, or the motor?

The second peak in microtubule fluorescence tends to be visible with the larger asters we form, and is near the boundary of the ‘core’ region of the aster. For our model, we exclude the core region since we do not believe the microtubules are well aligned in this region. We think that this second peak in fluorescence is due to the difference between the core region and the aster arms. In the core, the microtubules are likely not radially organized, since we do not observe alignment of the microtubules in that region via polarization microscopy, shown in Figure 1—figure supplement 4.

We added explanation of this point in lines 110-112, as follows:

“We tend to observe a 'shoulder' in the microtubule distribution (around 20*µ*m in the example in Figure 1(C)). This is around the size of the disordered aster core, which is discussed in SI section 8.11.”

Reviewer #3 (Recommendations for the authors):The authors succeed in demonstrating how the microscopic properties of motors can determine large-scale material structure. This is the putative principal impact of the work. However, some of these results have been seen in other motor/filament systems, such as actomyosin. An example is the relationship between contractile velocity and the dimension of the assembly (e.g. Thoresen, BJ, 2011, Linsmier et al., Nat Comm, 2016, Schuppler, Nat Comm, 2016). While similar relationships observed in microtubule /kinesin systems are notable, due to differences in motor proteins parameters, filament mechanics, etc, these prior works should be mentioned in light of the contributions of the present work.

We thank the reviewer for pointing out these references about telescoping contractility in actomyosin networks. We have added these references in lines 249-252, as follows:

“These findings are in agreement with results from several studies of contractile rates in actomyosin networks, that suggested 'telescoping' models of contraction, and suggest that this may be a common mechanism across cytosketetal networks [32-34].”